# Computational Aspects of Bayesian Persuasion under Approximate Best Response

**Kunhe Yang**
University of California, Berkeley
kunheyang@berkeley.edu

**Hanrui Zhang**
Chinese University of Hong Kong
hanrui@cse.cuhk.edu.hk

## Abstract

We study Bayesian persuasion under approximate best response, where the receiver may choose any action that is not too much suboptimal, given their posterior belief upon receiving the signal. We focus on the computational aspects of the problem, aiming to design algorithms that efficiently compute (almost) optimal strategies for the sender. Despite the absence of the revelation principle — which has been one of the most powerful tools in Bayesian persuasion — we design polynomial-time exact algorithms for the problem when either the state space or the action space is small, as well as a quasi-polynomial-time approximation scheme (QPTAS) for the general problem. On the negative side, we show there is no polynomial-time exact algorithm for the general problem unless P = NP. Our results build on several new algorithmic ideas, which might be useful in other principal-agent problems where robustness is desired.

## 1 Introduction

Bayesian persuasion [Kamenica and Gentzkow, 2011] concerns the problem where a principal (the "sender") incentivizes a self-interested agent (the "receiver") to act in certain ways by selectively revealing information about the state of the world. A commonly cited simplistic example — which nonetheless illustrates the essence of the problem — is that of selling apples.

**Example 1.1.** *Suppose a buyer (the receiver) is debating whether they should buy an apple from a seller (the sender). A priori, the buyer believes the apple (which is, say, a random apple from a large batch of apples) to be a "good" one with probability $1/3$, and a "bad" one with probability $2/3$. Moreover, suppose the buyer derives utility $1$ from buying a good apple, and $-1$ from buying a bad one. Then, aiming to maximize their expected utility, without any further information, the buyer should simply not buy the apple, because buying it would lead to expected utility $1 \times \frac{1}{3} + (-1) \times \frac{2}{3} = -\frac{1}{3} < 0$. The seller, who has perfect knowledge of the quality of the apple, and wants to "persuade" the buyer to buy the apple, can of course simply reveal the quality of the apple. Assuming the seller can do so in a credible way (which is a fundamental assumption in Bayesian persuasion), the buyer will buy the apple whenever it is good, which happens with probability $1/3$.*

*In fact, the seller can do even better by employing the following strategy: the seller promises to send a "signal" to the buyer regarding the quality of the apple, which can be either "probably good" or "definitely bad". More specifically, if the apple is good, the sender signals that it is "probably good"; if the apple is bad, the seller signals randomly, that it is "probably good" notwithstanding with probability $0.499$, and that it is "definitely bad" with probability $0.501$. Now consider the buyer's perspective, assuming the seller does act up to their promises. If the signal says "definitely bad", then the buyer should certainly not buy the apple. If, however, the signal says "probably good", then the buyer faces a more interesting decision. All the buyer knows is that the probability of (1) receiving "probably good" and (2) the apple is actually good, happening simultaneously, is $1/3$, and that the probability of (1) receiving "probably good" and (2) the apple is actually bad, happening*

*simultaneously, is $2/3 \times 0.499 < 1/3$. So, given the signal, the conditional probability that the apple is good is larger than $1/2$, which means the conditional expected utility of buying the apple is now positive. In other words, the seller has persuaded the rational buyer to buy the apple whenever the signal is "probably good", which happens with probability $1/3 + 2/3 \times 0.499 \approx 2/3$. This turns out to be (almost) the best that the seller can do.*

Since its introduction by Kamenica and Gentzkow [2011], Bayesian persuasion has attracted enormous attention from both economists and computer scientists. Indeed, a satisfactory solution to the problem is often twofold, involving an economic characterization that confines the sender's strategy space without loss of generality, and an efficient algorithm that finds the optimal strategy within this confined strategy space. For example, in the standard (and somewhat idealized) model of Bayesian persuasion, a prominent principle in economics, namely the revelation principle, states that without loss of generality, the sender's optimal strategy is to simply recommend an action (depending on the state of the world, effectively revealing partial information thereof) for the receiver to take, and make sure it is in the receiver's best interest to always follow such a recommendation, given their posterior belief of the state of the world. The computational problem of searching for the optimal strategy within this structured space of strategies turns out to be much easier than that of searching over the unconstrained strategy space. Such characterize-and-solve approaches have proved extremely successful in Bayesian persuasion — at least in settings where they apply.

The real world, unfortunately, is less precise than the idealized model. Even in the simplistic example of selling apples, there appear to be numerous subtleties that could potentially derail the supposedly (almost) optimal strategy of the seller. To name a few: the sender may not know exactly the receiver's prior belief, which might be different from the true distribution of the state of the world; the sender may not know exactly the receiver's utility function, which might be hard to evaluate even for the receiver themself; the device that generates randomness for the sender may be imperfect, affecting the receiver's posterior belief formed upon receiving a signal. In all these cases, the small inaccuracy can completely change the receiver's behavior: in the example of selling apples, the buyer would never buy the apple if they believe the apple is good a priori with probability $0.32$ instead of $1/3$, or if buying a good apple gives them utility $0.98$ instead of $1$, or if the device that generates randomness for the seller works in a way such that when the apple is bad, the seller actually signals "probably good" with probability $0.501$ instead of $0.499$. From the perspective of the sender, it would appear as if the receiver is acting in a somewhat suboptimal and unexpected way. As we will show later, the powerful revelation principle no longer applies in such cases, and the problem of finding the sender's optimal strategy becomes much trickier. On top of that, things become even more complicated when there are more than 2 possible states of the world, and/or more than 2 actions for the receiver to choose between. All this brings us to our main question: despite the lack of a structural characterization, is there a principled and efficient way of finding the optimal strategy for the sender that is robust against such inaccuracy?

## 1.1 Our Results and Techniques

In this paper, we study Bayesian persuasion in a natural model that accounts for the kind of subtleties discussed above: roughly speaking, the model allows the receiver to choose any action that is "not too much suboptimal", given their posterior belief upon receiving the signal. We focus our attention on computational aspects of the problem, aiming to design algorithms that efficiently compute (almost) optimal strategies.

**Direct-revelation signaling schemes are suboptimal.** Our first finding is negative: there exists extremely simple problem instances (with only 2 possible actions and 3 possible states), where any direct-revelation signaling scheme (i.e., a strategy of the sender where each possible signal is a recommendation of a single action, as discussed above) is suboptimal at least by a factor of 2, or an additive gap of $1/2$. This implies that the revelation principle ceases to work in our model, and the characterize-and-solve approach can no longer be employed.

**LP formulation, and efficient algorithm with small action spaces.** The above result highlights the need for new algorithmic ideas, and we present several of them in this paper. The first one is a linear program (LP) formulation for optimal strategies. The LP formulation has a similar high-level structure to the standard one for Bayesian persuasion: the variables correspond to probabilities that each signal is sent in each possible state of the world, and the constraints enforce the "semantics" of

the signals, in terms of how the receiver is expected to respond. However, one complication (among others) in our model is that multiple actions might be taken as the response to any fixed signal, so the semantics of a signal must be rich enough to encode the subset of actions that are possible in response to that signal (plus any additional information required to describe a signal). In particular, this means in general, the number of possible signals is exponential in the number of actions. As such, the LP formulation alone implies an efficient algorithm for our problem only when the number of actions is constant or logarithmic. Nonetheless, the LP formulation serves as a building block of our further algorithmic results.

**Efficient algorithm with small state spaces.** Next we design an efficient algorithm when the number of states is constant. To see how this is possible, observe that a subset of actions can each be chosen as the response to a signal, only if the posterior utilities corresponding to these actions are all close to the best possible. In other words, if there does not exist a posterior belief given which a subset of actions are all almost optimal, then this subset itself is "infeasible" as a set of possible responses. Such a subset of actions can never describe a signal in any strategy. Now the hope is to argue that the number of feasible subsets of actions is not too large. It turns out this can be obtained as a consequence of a result in combinatorial geometry, which bounds the number of "cells" in a low-dimensional space cut by a number of hyperplanes. We show that all feasible subsets induce a partition of these cells, so the same bound applies to the number of subsets too. In fact, one can show that the number of feasible subsets of actions is $n^{O(m)}$, where $n$ and $m$ are the number of actions and that of states, respectively.

Now we know the number of relevant subsets of actions cannot be too large, but it remains a problem to find these subsets. To this end, we make yet another geometric observation: the cells that correspond to feasible subsets of actions form a single "connected component", which means we can enumerate all feasible subsets by traversing this component. More specifically, we start from any feasible subset, and try all its "neighbors" by swapping in or out a single possible action. For each neighbor, we check its feasibility by solving another LP. By repeating this procedure we can reach all feasible subsets, in time polynomial in the number of feasible subsets. Once we have computed all feasible subsets, we simply solve the LP for optimal strategies restricted to these subsets, which is of polynomial size when the number of states $m$ is constant. This gives us an efficient algorithm with constant-size state spaces.

**Hardness of exact computation of the general problem.** Knowing that the problem can be solved with small action spaces or small state spaces, it is then natural to seek an efficient algorithm that works unconditionally, without restrictions on any parameters of the problem. We show, unfortunately, that such an algorithm does not exist unless $\mathsf{P} = \mathsf{NP}$. We do so by reducing from an "equally hard" variant of the subset sum problem: given a set of $2n$ integers that sum to $0$, decide whether there are $n$ integers out of the $2n$ that also sum to $0$. The idea of the reduction is that such a set of $n$ integers corresponds to a signal that gives the sender the highest posterior utility possible. To ensure this, the utility functions need to exhibit delicate structures, such that a signal that corresponds to either too many or too few integers must be suboptimal. The latter appears to be a quite ambivalent condition — in fact, this is only possible with the kind of inaccuracy that we consider. We believe the ideas of our reduction are potentially useful in other principal-agent problems where robustness is required.

**Approximation in quasi-polynomial time.** In light of the hardness result, we turn our attention to approximation algorithms. We present a quasi-polynomial-time approximation scheme (QPTAS): for any target additive error $\varepsilon > 0$, we give an $\varepsilon$-approximate algorithm that runs in time quasi-polynomial in $m$ and $n$, where the time complexity may depend on $\varepsilon$. The idea is to cover the space of all possible posterior utility functions using small cells, and consider a representative point in each cell with a small error. It is known that a covering of size $O(\log n/\varepsilon^2)$ exists, which has already proven useful in other game-theoretic computational problems [Althöfer, 1994, Lipton et al., 2003, Gan et al., 2023]. However, one difficulty in our model is that there are certain types of errors that are unacceptable, no matter how small. For example, fixing a strategy, the sender's utility can be discontinuous in the receiver's utility function, and any error near points of discontinuity can lead to a huge gap in the sender's utility. Similarly, because of the robustness component in the model, a small error in the sender's utility may lead to a totally different response from the receiver. To avoid such gaps, we allow approximation only in the sender's utility, and refrain from estimating the action that the receiver will respond with. Instead, we consider the approximate worst-case utility of the

sender, which turns out to be tractable using linear constraints involving approximate utility functions. Combining this with the LP formulation discussed above, we have an LP of quasi-polynomial size, which can be solved in quasi-polynomial time.

## 1.2 Related Works

We defer a detailed discussion of related work to Appendix A. Our work connects to two lines of work, the computational aspect of Bayesian persuasion [Dughmi and Xu, 2016, Dughmi, 2017, 2014, Bhaskar et al., 2016, Rubinstein, 2017, Babichenko and Barman, 2016, 2017, Xu, 2020, Zhou et al., 2022] and various notions of robust Bayesian persuasion [de Clippel and Zhang, 2022, Chen and Lin, 2023, Camara et al., 2020, Zu et al., 2021, Collina et al., 2023, Kosterina, 2022, Dworczak and Pavan, 2022, Hu and Weng, 2021, Castiglioni et al., 2020, Wu et al., 2022, Babichenko et al., 2022]. We also discuss the comparison between our paper and that of Gan et al. [2023] on robust Stackelberg games.

## 2 Preliminaries

Our model builds on the classic Bayesian persuasion model with a single sender and a single receiver. At a high level, the model formalizes a scenario where the sender, possessing private information about the true state, aims to influence the receiver's decision-making by strategically sending partial information about the true state according to a pre-committed signaling scheme. We start by introducing the basic setting and notations, and then introduce the robust Bayesian persuasion model that considers a receiver who acts not exactly, but approximately in accordance with their best interest in the decision-making process.

### 2.1 Bayesian persuasion: the classical model

**Basic setting and notations.** Let $\Omega$ be the states of the world with $|\Omega| = m$, and $\Delta(\Omega)$ be the set of all probability distributions on $\Omega$. For all distributions $\mu \in \Delta(\Omega)$, let $\mathsf{supp}(\mu)$ be the support of $\mu$, i.e., $\mathsf{supp}(\mu) \triangleq \{\omega \in \Omega \mid \mu(\omega) > 0\}$. We use $\mu_0 \in \Delta(\Omega)$ to denote the prior distribution over states and assume that $\mu_0$ is common knowledge between the sender and the receiver.

**Signaling scheme.** Let $\Sigma$ ($|\Sigma| < \infty$) be a finite set of signals. A *signaling scheme* $\varphi : \Omega \to \Sigma$ is a randomized mapping from the states of the world to probability distributions over signals. In the Bayesian persuasion protocol, the sender first commits to a signaling scheme $\varphi$, then observes the true state of the world $\omega \sim \mu_0$, and sends signal $\sigma \sim \varphi(\omega)$ to the receiver. We use $\varphi(\omega, \sigma)$ to denote the probability of sending signal $\sigma$ conditioning on observing $\omega$ as the true state, and $\varphi(\sigma) = \sum_{\omega' \in \Omega} \mu_0(\omega')\varphi(\omega', \sigma)$ to denote the marginal probability of a signal $\sigma \in \Sigma$ being realized.

Upon receiving the signal $\sigma$, the receiver performs a Bayes update on $\mu_0$ using knowledge about the scheme $\pi$ to obtain a posterior belief $\mu_\sigma \in \Delta(\Omega)$, i.e.,

$$\mu_\sigma(\omega) = \frac{\mu_0(\omega)\varphi(\omega, \sigma)}{\varphi(\sigma)}.$$

In addition, algebraic manipulations based on the Bayes' rule suggest that the signaling scheme $\varphi$ can be viewed as the process of creating a distribution over posterior distributions that satisfy the Bayes plausibility condition [Kamenica and Gentzkow, 2011, Gentzkow and Kamenica, 2016]:

$$\forall \omega \in \Omega, \qquad \mu_0(\omega) = \sum_{\omega \in \Omega} \varphi(\sigma) \cdot \mu_\sigma(\omega). \qquad \text{(Bayes plausibility)}$$

Throughout this paper, we will frequently adopt this perspective, especially in the characterization of (robust) utilities.

**Utilities and best response sets** Although only the receiver can take actions, their action influences both the sender and the receiver's utility, both of which also depend on the state of the world. Let $\mathcal{A}$ be the action space of the receiver with $|\mathcal{A}| = n$. We use $s : \Omega \times \mathcal{A} \to [0, 1]$ to denote the sender's utility and $r : \Omega \times \mathcal{A} \to [0, 1]$ to denote the receiver's utility, where both utilities are normalized to be between 0 and 1. Additionally, for distributions $\mu \in \Delta(\Omega)$, we abuse the notations and use $s(\mu, a) = \mathbb{E}_{\omega \sim \mu} s(\omega, a)$ and $r(\mu, a) = \mathbb{E}_{\omega \sim \mu} r(\omega, a)$ to denote the sender and receiver's expected utilities under the state distribution $\mu$.

**Receiver's strategies**  A receiver's strategy, denoted with $\rho : \Sigma \to \mathcal{A}$, is a (possibly randomized) mapping from the observed signals to actions which specifies the receiver's strategy of choosing responses. Given a signaling scheme $\varphi$ and a receiver strategy $\rho$, we use $S(\varphi, \rho)$ to denote the expected sender utility, which is computed as

$$S(\varphi, \rho) \triangleq \sum_{\sigma \in \Sigma} \varphi(\sigma) \cdot s(\mu_\sigma, \rho(\sigma)) = \sum_{\omega \in \Omega} \sum_{\sigma \in \Sigma} \mu_0(\omega) \varphi(\omega, \sigma) s(\omega, \rho(\sigma)).$$

The classical model of Bayesian persuasion assumes that the receiver's strategy $\rho(\sigma)$ is always the exact best response of the posterior distribution $\mu_\sigma$ that maximizes their expected posterior utility (ties are broken in favor of the sender). Formally, the best response action action is defined as

$$a^\star(\mu_\sigma) \triangleq \underset{a \in \mathcal{A}}{\operatorname{argmax}}\, r(\mu_\sigma, a),\, {}^1$$

and the receiver's strategy is assumed to satisfy $\rho(\sigma) = a^\star(\mu_\sigma)$. In this paper, we will relax this exact best-response assumption to allow for approximate best responses.

## 2.2   Robust Bayesian persuasion with approximate best responses

In this paper, we allow for some degree of suboptimality in the receiver's response. Specifically, we assume that the expected utility under receiver's response $\rho(\sigma)$ is not too suboptimal compared to the best action $a^\star(\mu_\sigma)$. Formally, if we use $\mathsf{BR}_\delta(\cdot)$ to denote the set of $\delta$-optimal responses on the input belief:

$$\mathsf{BR}_\delta(\mu) \triangleq \{a \in \mathcal{A} \mid r(\mu, a) > r(\mu, a^\star(\mu)) - \delta\}, \tag{1}$$

then a receiver's strategy $\rho : \Sigma \to \mathcal{A}$ is $\delta$-best response (or $\delta$-BR) if it satisfies $\rho(\sigma) \in \mathsf{BR}_\delta(\mu_\sigma)$ for all $\sigma \in \Sigma$. We use $\mathfrak{BR}_\delta(\varphi)$ to denote the set of all $\delta$-BR strategies under the signaling scheme $\varphi$.

**Robust sender utility**  In this paper, we adopt the max-min adversarial robustness perspective and aim to offer robustness guarantees against worst-case $\delta$-BR strategies. We use the *robust utility* $\widehat{S}_\delta$ of a signaling scheme to characterize the expected utility under the worst-case $\delta$-BR strategy:

$$\widehat{S}_\delta(\varphi) = \min_{\rho_\delta \in \mathfrak{BR}_\delta(\varphi)} S(\varphi, \rho_\delta).$$

The sender aims to maximize the robust utility through optimizing the signaling scheme. Equivalently, the sender faces a bi-level optimization problem of the following max-min form:

$$\widehat{S}_\delta^\star \triangleq \max_{\varphi^\star} \widehat{S}_\delta(\varphi^\star) = \max_{\varphi^\star} \min_{\rho_\delta \in \mathfrak{BR}_\delta(\varphi^\star)} S(\varphi^\star, \rho_\delta). \qquad \text{(sender's objective)}$$

Solving the optimization problem in (sender's objective) includes (1) finding an appropriate signal space $\Sigma$ that is sufficient to achieve the optimal utility and (2) optimizing for the optimal signaling scheme given the signal space.

**Remark 2.1** (worst-case $\delta$-BR strategy). *It is not hard to see that the worst-case $\delta$-BR strategy $\rho_\delta$ that achieves* (sender's objective) *will choose the worst action that minimizes the sender's utility for each posterior distribution separately, i.e.,*

$$\rho_\delta(\sigma) \in \underset{a \in \mathsf{BR}_\delta(\mu_\sigma)}{\operatorname{argmin}}\, s(\mu_\sigma, a).$$

*Therefore, the robust utility under any signaling scheme $\varphi$ can be equivalently written as*

$$\widehat{S}_\delta(\varphi) = \sum_{\sigma \in \Sigma} \varphi(\sigma) \cdot s(\mu_\sigma, \rho_\delta(\sigma)) = \sum_{\sigma \in \Sigma} \varphi(\sigma) \cdot \min_{a \in \mathsf{BR}_\delta(\mu_\sigma)} s(\mu_\sigma, a).$$

**Remark 2.2** (Strict inequality in definition of $\mathsf{BR}_\delta$ set). *In eq. (1), we use strict inequality so that the sender's strategy space is closed and compact, and the sender's objective is well-defined. This is a non-essential choice: one could instead define $\delta$-BR responses using weak inequality and investigate the sender's* supremum *robust utility, which would not change the nature of the problem.*

---

[1]When there are multiple actions $a^\star(\mu_\sigma)$ that maximizes the receiver's utility, we define $a^\star(\mu_\sigma)$ to be the maximizer that achieves the highest sender's utility.

# 3 LP formulation and algorithm with small action spaces

As we show in Appendix B, the powerful revelation principle fails in the robustness model that we study. In fact, we establish the following claim.

**Proposition 3.1** (Proposition B.1 in Appendix B). *There exists a sequence of Bayesian persuasion instances with a robustness level $\delta = \Theta(1)$, such that the following holds in the limit: any direct-revelation scheme $\varphi$ is suboptimal, at least by a factor of $2$ or an additive gap of $\frac{1}{2}$. That is, for any direct-revelation scheme $\varphi$,*

$$\widehat{S}_\delta(\varphi) \leq \frac{1}{2}\widehat{S}_\delta^\star, \quad and \quad \widehat{S}_\delta(\varphi) \leq \widehat{S}_\delta^\star - \frac{1}{2}.$$

The above proposition highlights the need for new algorithmic ideas to efficiently compute (approximately) optimal robust schemes.

In this section, we present a linear program (LP) that computes the optimal robust utility $\widehat{S}_\delta^\star$ and the optimal robust signaling scheme $\varphi^\star$. Although the high-level idea is similar to the LP formulation for computing the optimal non-robust scheme in standard Bayesian persuasion [Dughmi and Xu, 2016, Dughmi, 2017], our robust LP formulation follows significantly different semantics.

We characterize the maximum number of signals needed to achieve the optimal robust utility in Lemma 3.2. This lemma can be viewed as a generalized (and unfortunately, much less powerful due to lack of best responses) version of the revelation-principle style argument in [Kamenica and Gentzkow, 2011, Proposition 1] that accounts for worst-case approximate best responses. See Appendix C for the proof of the lemma.

**Lemma 3.2.** *There exists an optimal robust signaling scheme that is supported on at most $n \cdot 2^{n-1}$ signals, in which each signal $\sigma \in \Sigma$ corresponds to a unique pair of $(A, \tilde{a})$ such that $\tilde{a} = a^\star(\mu_\sigma)$ and $A = \mathsf{BR}_\delta(\mu_\sigma)$.*

According to Lemma 3.2, it suffices to consider the following signal space:

$$\Sigma = \{(A, \tilde{a}) \mid A \subseteq \mathcal{A}, \tilde{a} \in A\},$$

where each signal $\sigma = (A, \tilde{a})$ satisfies both $a^\star(\mu_\sigma) = \tilde{a}$ and $\mathsf{BR}_\delta(\mu_\sigma) = A$. Recall from Remark 2.1 that the robust utility can be written as

$$\widehat{S}_\delta^\star = \max_\varphi \sum_{(A,\tilde{a})\in\Sigma} \varphi((A,\tilde{a})) \cdot \min_{a\in\mathsf{BR}_\delta(\mu_{(A,\tilde{a})})} s(\mu_{(A,\tilde{a})}, a).$$

Translating this max-min optimization to a constrained maximization problem, and enforcing the abovementioned semantics for each signal, we arrive at the following (non-linear) program in Fig. 1. Although the program in Fig. 1 is not yet a linear program, both the first and second constraints can

---

$$
\begin{aligned}
\underset{\boldsymbol{\varphi},\boldsymbol{x}}{\text{maximize}} \quad & \sum_{(A,\tilde{a})\in\Sigma} \boldsymbol{x}(A,\tilde{a}) \\
\text{subject to} \quad & \boldsymbol{\varphi}(\mu_{(A,\tilde{a})}) \cdot s(\mu_{(A,\tilde{a})}, a) \geq \boldsymbol{x}(A,\tilde{a}). && \forall \sigma \in \Sigma, \forall a \in A \\
& a^\star(\mu_\sigma) = \tilde{a} && \forall (A,\tilde{a}) \in \Sigma \\
& \mathsf{BR}_\delta(\mu_{(A,\tilde{a})}) = A && \forall (A,\tilde{a}) \in \Sigma
\end{aligned}
$$

Figure 1: A program for the optimal robust signaling scheme supported on $\Sigma = \{(A, \tilde{a})\}$.

---

be equivalently written as linear constraints on the conditional probabilities $\boldsymbol{\varphi}(\omega, (A, \tilde{a}))$ that define the signaling scheme $\varphi$, using the following observation based on the Bayes' rule:

$$\forall \sigma \in \Sigma, \quad \boldsymbol{\varphi}(\mu_\sigma) \cdot s(\mu_\sigma, a) = \sum_{\omega\in\Omega} \mu_0(\omega)\boldsymbol{\varphi}(\omega, \sigma) \cdot s(\omega, a). \tag{2}$$

Therefore, it remains to characterize the $\delta$-BR response sets, and would be ideal if they could also be equivalently expressed as linear constraints. Unfortunately, the definition of $\delta$-BR sets involves strict-inequality constraints due to the issues discussed in Remark 2.2. In particular, for each $(A, \tilde{a}) \in \Sigma$, the third constraint is equivalent to

    (1) $\forall a \in A, \ s(\mu_\sigma, a) > s(\mu_\sigma, \tilde{a}) - \delta$;    (2) $\forall a \in \mathcal{A} \setminus A, \ s(\mu_\sigma, a) \leq s(\mu_\sigma, \tilde{a}) - \delta$,

where (1) involves a strict inequality and could lead to open polytopes. Due to such subtleties, we first consider the LP in Fig. 2 which is a relaxation of the program in Fig. 1 by dropping the strict-inequality constraint (1). The last two constraints in Fig. 1 guarantee that variables $\varphi(\omega, A, \tilde{a})$ give rise to a valid signal distribution for every state $\omega \in \Omega$.

$$
\begin{aligned}
\underset{\varphi, \boldsymbol{x}}{\text{maximize}} \quad & \sum_{(A,\tilde{a}) \in \Sigma} \boldsymbol{x}(A, \tilde{a}) \\
\text{subject to} \quad & \sum_{\omega \in \Omega} \mu_0(\omega)\varphi(\omega, A, \tilde{a})s(\omega, a) \geq \boldsymbol{x}(A, \tilde{a}), && \forall (A, \tilde{a}) \in \Sigma, \forall a \in A \\
& \sum_{\omega \in \Omega} \mu_0(\omega)\varphi(\omega, A, \tilde{a}) \left(r(\omega, \tilde{a}) - r(\omega, a)\right) \geq 0, && \forall (A, \tilde{a}) \in \Sigma, \forall a \in A \\
& \sum_{\omega \in \Omega} \mu_0(\omega)\varphi(\omega, A, \tilde{a}) \left(r(\omega, \tilde{a}) - r(\omega, a) - \delta\right) \leq 0, && \forall (A, \tilde{a}) \in \Sigma, \forall a \in \mathcal{A} \backslash A \\
& \sum_{(A,\tilde{a}) \in \Sigma} \varphi(\omega, A, \tilde{a}) = 1 && \forall \omega \in \Omega \\
& \varphi(\omega, A, \tilde{a}) \geq 0. && \forall (A, \tilde{a}) \in \Sigma
\end{aligned}
$$

Figure 2: Relaxed LP for the optimal robust signaling scheme

Since Fig. 2 is a relaxation of the original program, its optimal objective provides an upper bound for $\widehat{S}_\delta^\star$. However, as we will show later, not only does the optimal objective value exactly equal $\widehat{S}_\delta^\star$, but the optimal variables $\varphi^\star$ also exactly characterize the optimal robust signaling scheme that achieves this $\widehat{S}_\delta^\star$. This gives us an algorithm that efficiently computes the optimal robust signaling scheme when the number of action spaces is constant or polynomial. We formalize this in Proposition 3.3, and provide a proof in Appendix C.

**Proposition 3.3** (Efficient algorithm for small action space). *The optimal robust signaling scheme can be computed by the linear program in Fig. 2 with size $O(2^n mn)$.*

## 4 Efficient algorithm with small state spaces

In this section, we focus on the robust persuasion instances where the action space is large, but the state space is small. For such instances, our key observation is that among the $(n-1)2^n$ tuples in $\Sigma$, only a small fraction of them can be realized by posterior distributions and therefore serve as feasible signal candidates. In Section 4.1, we characterize the structural properties of feasible tuples $(A, \tilde{a}) \in \Sigma$ and draw connection to the polytopes in the simplex supported on the state space. In Section 4.2, we leverage these structural insights and design an efficient algorithm that accelerates the computation of the optimal robust signaling scheme.

### 4.1 Structural properties

We formally define the feasibility of a candidate tuple $(A, \tilde{a}) \in \Sigma$ according to the existence of a posterior distribution that has $A$ as its $\delta$-BR set and $\tilde{a}$ as its best response action.

**Definition 4.1** (Feasible subset-action tuple). *A tuple $(A, \tilde{a}) \in \Sigma$ is feasible if there exists some posterior distribution $\mu \in \Delta(\Omega)$ such that $a^\star(\mu) = \tilde{a}$ and $\mathsf{BR}_\delta(\mu) = A$. We use $\Sigma^\dagger$ to denote the set of all feasible tuples in $\Sigma$.*

For each feasible tuple $(A, \tilde{a}) \in \Sigma^\dagger$, let $\Delta_{(A,\tilde{a})}$ be the set of posterior distributions that satisfy both constraints in Definition 4.1:

$$
\Delta_{(A,\tilde{a})} \triangleq \{\mu \in \Delta(\Omega) \mid a^\star(\mu) = \tilde{a}, \ \mathsf{BR}_\delta(\mu) = A\}
$$

It is not hard to see that the subsets $\Delta_{(A,\tilde{a})}$ partitions the simplex, because each distribution in the simplex is associated with a unique best response action and $\delta$-BR set. That is,

$$
\Delta(\Omega) = \bigcup_{(A,\tilde{a}) \in \Sigma^\dagger} \Delta_{(A,\tilde{a})}.
$$

In addition, each $\Delta_{(A,\tilde{a})}$ is a polytope in the simplex that is defined by the following constraints that are linear in the distribution $\mu$:

$$\mu \in \Delta_{(A,\tilde{a})} \iff \begin{cases} r(\mu, a) \le r(\mu, \tilde{a}), & \forall a \in A; \\ r(\mu, a) > r(\mu, \tilde{a}) - \delta, & \forall a \in A; \\ r(\mu, a) \le r(\mu, \tilde{a}) - \delta, & \forall a \in \mathcal{A} \setminus a. \end{cases} \tag{3}$$

Our key observation is that the linear constraints that characterize different $\Delta_{(A,\tilde{a})}$ all take one of the following two forms based on an ordered pair of actions $(a, a') \in \mathcal{A} \times \mathcal{A}$:

1. Either $r(\mu, a) \le r(\mu, a') - \delta$, or its complement $r(\mu, a) > r(\mu, a') - \delta$;
2. $r(\mu, a) \le r(\mu, a')$,

which give rise to no more than $2n^2$ hyperplanes. Although the number of hyperplanes scales with the potentially large number of actions, when the state space is small, these hyperplanes, and the polytopes they define, all live within a low-dimensional space. According to a fundamental theorem in computational geometry (see, e.g., Theorem 28.1.1 in [Halperin and Sharir, 2017]), these hyperplanes cut the $m$-dimensional into at most $O\left((2n^2)^{m-1}\right)$ cells. This observation helps us to bound the number of feasible tuples in the following lemma. The proof of Lemma 4.2 is deferred to Appendix D.2.

**Lemma 4.2.** *The size of the feasible tuples $\Sigma^{\dagger}$ satisfy $|\Sigma^{\dagger}| \le \min\left\{n2^{n-1}, n^{O(m)}\right\}$.*

Next, we establish some structural properties on the set of feasible tuples $\Sigma^{\dagger}$ that could facilitate the efficient search over candidate signals. In the remainder of this section, we first define a bounded-degree *symmetric difference graph* on all tuples in $\Sigma$, then show that all feasible tuples in $\Sigma^{\dagger}$ form a connected component in the symmetric difference graph.

**Definition 4.3** (Symmetric difference graph). *In the symmetric difference graph $G(\Sigma, E)$, each vertex is a tuple $(A, \tilde{a}) \in \Sigma$. Each edge $\{(A_1, \tilde{a}_1), (A_2, \tilde{a}_2)\} \in E$ represents that the symmetric difference between tuples $(A_1, \tilde{a}_1)$ and $(A_2, \tilde{a}_2)$ is of size 1, which is satisfied for the following two cases:*

- *$A_1 = A_2$, and $\tilde{a}_1 \ne \tilde{a}_2$.*
- *$\tilde{a}_1 = \tilde{a}_2 \in A_1 \cap A_2$, and $|(A_1 \setminus A_2) \cup (A_2 \setminus A_1)| = 1$.*

The symmetric difference graph $G$ characterizes the structural relationship between tuples in $\Sigma$, in which only any two tuples are connected if and only if their symmetric difference is small enough. Together with the following lemma, $G$ provides a useful framework for us to search within the exponentially large vertex set.

**Lemma 4.4** (Connectivity). *The subgraph of $G$ induced by $\Sigma^{\dagger}$ is connected.*

We defer the proof of Lemma 4.4 to Appendix D.1. At a high level, the proof translates the connectivity of the polytopes $\Delta_{(A,\tilde{a})}$ in the simplex space to the connectivity of feasible tuples in the symmetric distance graph $G$ by analyzing the geometric properties of the common face of shared by every pair of adjacent polytopes.

### 4.2 Algorithm

In this section, we present Algorithm 1 that leverages the connectivity of the feasible tuples to efficiently search for $\Sigma^{\dagger}$. This algorithm essentially performs depth-first-search (DFS) on the connectivity graph by recursively searching all the neighbors with symmetric difference of size 1 to the given tuple that it's currently searching. The initial tuple to start the EXPLORE procedure can be set to any tuple that is already known to be feasible, e.g., the $\delta$-BR set and the optimal response associated with the prior distribution $\mu_0$.

Note that each tuple $(A, \tilde{a}) \in \Sigma$ is feasible if and only if there exists $\mu \in \Sigma$ that satisfies all three constraints in Equation (3) simultaneously, which can be written as checking the feasibility of the corresponding LP (with strict-inequality constraints) on the distributions $\boldsymbol{\mu}$. To replace strict-inequality constraints with non-strict constraints, we propose to check the LP via a margin-maximization trick in Fig. 4 of Appendix D.3. This LP is defined on a closed polytope, and asserts feasibility if and only if the optimal margin is strictly positive, i.e., $\varepsilon^{\star} > 0$.

---

**Algorithm 1:** EXPLORE

---

**Input:** tuple $(A, \tilde{a}) \in \Sigma$

1  Check the feasibility of $(A, \tilde{a})$ by solving the LP in Fig. 4, $\varepsilon^\star \leftarrow$ optimal objective value;

2  **if** $\varepsilon^\star > 0$ **then**

3       Mark $(A, \tilde{a})$ checked and feasible;

4       **for** *actions* $a \in A \setminus \{\tilde{a}\}$ **do**

5           If $(A \setminus \{a\}, \tilde{a})$ is unchecked, call EXPLORE$((A \setminus \{a\}, \tilde{a}))$;

6           If $(A, a)$ is unchecked, call EXPLORE$((A, a))$;

7       **end**

8       **for** *actions* $a \in \mathcal{A} \setminus A$ **do**

9           If $(A \cup \{a\}, \tilde{a})$ is unchecked, call EXPLORE$((A \cup \{a\}, \tilde{a}))$;

10      **end**

11 **end**

12 **else**

13      Mark $(A, \tilde{a})$ checked and infeasible

14 **end**

---

**Theorem 4.5.** *Running Algorithm 2 with initial tuple* $(\mathsf{BR}_\delta(\mu_0), a^\star(\mu_0))$ *finds all feasible tuples* $\Sigma^\dagger \subseteq \Sigma$ *by solving at most* $n^{O(m)}$ *LPs, each of size* $O(m + n)$.

*Proof.* The correctness of this algorithm is ensured by the connectivity property in Lemma 4.4. It remains to upper bound the number of feasibility checks performed. Since Algorithm 1 only checks new a vertex when it has not been checked before, the total number of feasibility checks is upper bounded by the size of the closed neighborhood of $\Sigma^\dagger$. Since each tuple $(A, \tilde{a}) \in \Sigma$ has at most $O(n)$ neighbors, the degree of any vertex in the symmetric difference graph $G$ is upper bounded by $O(n)$. Therefore, the size of the closed neighborhood is at most $O(n) \cdot |\Sigma^\dagger| \leq n^{O(m)}$. $\qquad\qquad\square$

The final step for optimizing signaling schemes is to solve the LP in Fig. 2 with the original signal space $\Sigma$ replaced by feasible tuples $\Sigma^\dagger$. We summarize the entire procedure and its complexity in Algorithm 2 and Corollary 4.6.

**Corollary 4.6.** *When the number of states* $m$ *is small, the optimal signaling scheme can be efficiently computed by Algorithm 2, which involves solving* $n^{O(m)}$ *LPs of size* $O(m + n)$ *and then solve a single LP of size* $m \cdot n^{O(m)}$.

---

**Algorithm 2:** ALGORITHM FOR SMALL STATE SPACES

---

$(A_0, \tilde{a}_0) \leftarrow (\mathsf{BR}_\delta(\mu_0), a^\star(\mu_0))$;

Search for $\Sigma^\dagger$ by running SEARCH$(A_0, \tilde{a}_0)$;

Solve the LP in Fig. 2 with $\Sigma$ replaced by $\Sigma^\dagger$.

---

## 5 Approximation algorithm for the general problem

In this section, we first show that the general problem without any restrictions is computationally hard, then present a quasi-polynomial time approximation scheme (QPTAS) for the problem. The following claim (proof defered to Appendix E) establishes the computational hardness of finding the exact optimal robust signaling scheme. This result implies that a polynomial-time exact algorithm for the robust persuasion problem does not exist unless $\mathsf{P} = \mathsf{NP}$.

**Theorem 5.1.** *The problem of computing the exact optimal robust utility is* $\mathsf{NP}$*-hard.*

In the following, we present a QPTAS that computes an $\varepsilon$-approximate optimal $\delta$-robust signaling scheme $\widehat{\varphi}$ for any given $\delta > 0$, such that

$$\widehat{S}_\delta(\widehat{\varphi}) \geq \widehat{S}_\delta^\star - \varepsilon.$$

Our $\delta$-optimal signaling scheme is supported on a significantly different signal space compared to the optimal schemes for cases of small state space and small action space. The high-level idea is, instead of enumerating each $(\mathsf{BR}_\delta(\mu), a^\star(\mu))$ set, we partition the simplex $\Delta(\Omega)$ into small cells, each centered around a $k$-uniform distribution $\overline{\mu}$ (to be defined later), and incorporate additional semantics to ensure that the utility profiles $s(\overline{\mu}, \cdot)$ are representative enough for all distributions within this cell. It is useful to have the utilities at $\overline{\mu}$ being representative because we can then determine the $\delta$-BR sets according to the relative magnitude of utilities at $\overline{\mu}$ and translate the requirements into linear constraints.

Formally, for an integer $k$, a distribution $\mu \in \Delta(\Omega)$ is $k$-uniform if $\forall \omega \in \Omega$, $\mu(\omega) = \frac{k_\omega}{k}$ for some integer $k_\omega \le k$. Let $\mathcal{G}_k \subseteq \Delta(\Omega)$ denote the set of all $k$-uniform distributions on the state space. We have $|\mathcal{G}_k| = O(m^k)$. Moreover, by [Althöfer, 1994], for $k_\varepsilon = \left\lceil \frac{\log(2n)}{2\varepsilon^2} \right\rceil$, we have:

$$\forall \mu \in \Delta(\Omega), \ \exists \overline{\mu} \in \mathcal{G}_{k_\varepsilon}, \ \text{s.t.} \qquad \forall a \in \mathcal{A}, \ |s(\mu, a) - s(\overline{\mu}, a)| \le \varepsilon.$$

Accordingly, we define the $\varepsilon$-cell around $\overline{\mu} \in \mathcal{G}_k$ to be

$$\mathcal{C}_\varepsilon^{\overline{\mu}} = \left\{ \mu \in \Delta(\Omega) : \ \forall a \in \mathcal{A}, \ |s(\mu, a) - s(\overline{\mu}, a)| \le \varepsilon \right\}.$$

Note that given $\varepsilon$ and $\overline{\mu}$, $\mu \in \mathcal{C}_\varepsilon^{\overline{\mu}}$ is equivalent to a set of $2n$ linear constraints on $\mu$.

We consider the following signal space:

$$\widehat{\Sigma} = \left\{ (\overline{\mu}, \tilde{a}, \dot{a}) \mid \overline{\mu} \in \mathcal{G}_{k_{\varepsilon'}}, \ \tilde{a}, \dot{a} \in A \right\}, \qquad \text{where } k_{\varepsilon'} = \left\lceil \frac{\log(2n)}{2(\varepsilon')^2} \right\rceil, \ \varepsilon' = \frac{\varepsilon}{5} \tag{4}$$

Define LP variables $\varphi(\omega, \overline{\mu}, \tilde{a}, \dot{a})$ for each $\omega \in \Omega$ and each signal $(\overline{\mu}, \tilde{a}, \dot{a}) \in \widehat{\Sigma}$ to represent the conditional probabilities of sending that signal. Each signal $\sigma = (\overline{\mu}, \tilde{a}, \dot{a})$ is specified by the $k$-uniform distribution $\overline{\mu}$ that centers the cell, the best response action $\tilde{a}$, and the action $\dot{a}$ in $\delta$-BR set that minimizes the sender's strategy. The LP is defined in Fig. 3.

$$\begin{aligned}
&\underset{\varphi, x}{\text{maximize}} && \sum_{\omega \in \Omega} \mu_0(\omega) \sum_{(\overline{\mu}, \tilde{a}, \dot{a}) \in \widehat{\Sigma}} \varphi(\omega, \overline{\mu}, \tilde{a}, \dot{a}) s(\omega, \dot{a}) \\
&\text{subject to} && \sum_{\omega \in \Omega} \mu_0(\omega) \varphi(\omega, \overline{\mu}, \tilde{a}, \dot{a}) \cdot \left( s(\omega, a) - s(\overline{\mu}, a) + \varepsilon' \right) \ge 0 \quad (\forall a \in \mathcal{A}, \ \forall (\overline{\mu}, \tilde{a}, \dot{a}) \in \widehat{\Sigma}) \\
& && \sum_{\omega \in \Omega} \mu_0(\omega) \varphi(\omega, \overline{\mu}, \tilde{a}, \dot{a}) \cdot \left( s(\omega, a) - s(\overline{\mu}, a) - \varepsilon' \right) \le 0 \quad (\forall a \in \mathcal{A}, \ \forall (\overline{\mu}, \tilde{a}, \dot{a}) \in \widehat{\Sigma}) \\
& && \sum_{\omega \in \Omega} \mu_0(\omega) \varphi(\omega, \overline{\mu}, \tilde{a}, \dot{a}) \left( r(\omega, \tilde{a}) - r(\omega, a) \right) \ge 0, \qquad (\forall a \in \mathcal{A}, \ \forall (\overline{\mu}, \tilde{a}, \dot{a}) \in \widehat{\Sigma}) \\
& && \sum_{\omega \in \Omega} \mu_0(\omega) \varphi(\omega, \overline{\mu}, \tilde{a}, \dot{a}) \left( r(\omega, \tilde{a}) - r(\omega, a) - \delta \right) \ge 0, \\
& && \hspace{6cm} (\forall a \ \text{s.t.} \ s(\overline{\mu}, a) < s(\overline{\mu}, \dot{a}) - 2\varepsilon') \\
& && \hspace{6cm} (\forall (\overline{\mu}, \tilde{a}, \dot{a}) \in \widehat{\Sigma}) \\
& && \sum_{(\overline{\mu}, \tilde{a}, \dot{a}) \in \widehat{\Sigma}} \varphi(\omega, \overline{\mu}, \tilde{a}, \dot{a}) = 1 \hspace{4cm} (\forall \omega \in \Omega) \\
& && \varphi(\omega, \overline{\mu}, \tilde{a}, \dot{a}) \ge 0. \hspace{4cm} (\forall \omega \in \Omega, \forall (\overline{\mu}, \tilde{a}, \dot{a}) \in \widehat{\Sigma})
\end{aligned}$$

Figure 3: QPTAS for computing an $\varepsilon$-approximate robust signaling scheme, where $\varepsilon' = \frac{\varepsilon}{5}$.

Intuitively, the first two constraints in Fig. 3 are designed to ensure $\mu_{(\overline{\mu}, \tilde{a}, \dot{a})} \in \mathcal{C}_\varepsilon^{\overline{\mu}}$. The third and fourth constraints describe the semantics of the receiver's response strategy. Finally, the last two constraints guarantee that $\varphi$ represents a valid signal distribution on each state. We summarize the guarantee of the QPTAS in Theorem 5.2 and formally prove it in appendix F.

**Theorem 5.2.** *For any $\varepsilon > 0$ and $\delta > 0$, the optimal solution to the LP in Fig. 3 is an $\varepsilon$-approximate $\delta$-robust signaling scheme that is supported on $O\left( n^2 m^{\lceil 12.5 \log(2n)/(\varepsilon^2) \rceil} \right)$ signals.*

**Acknowledgments**

This material is based upon work supported by the National Science Foundation under Grant No. DMS-1928930 and by the Alfred P. Sloan Foundation under grant G-2021-16778, while Hanrui Zhang was in residence at the Simons Laufer Mathematical Sciences Institute (formerly MSRI) in Berkeley, California, during the Fall 2023 semester.

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

## A    Related Works

**Bayesian persuasion.**    Our paper generalizes the canonical model of Bayesian persuasion for information design that was introduced by the seminal work of Aumann et al. [1995], Kamenica and Gentzkow [2011], Brocas and Carrillo [2007]. This framework has been extended to multi-receiver settings [Bergemann and Morris, 2016, 2019] and supports many important applications, such as voting [Schnakenberg, 2015, Alonso and Câmara, 2016, Bardhi and Guo, 2018, Wang, 2013, Arieli and Babichenko, 2019], bilateral trades [Bergemann et al., 2015, Bergemann and Pesendorfer, 2007], incentivizing exploration [Mansour et al., 2020, 2022, Kremer et al., 2014], to name a few. We refer the readers to survey papers [Candogan, 2020, Kamenica, 2019] for a comprehensive overview.

**Computational aspects of Bayesian persuasion and information design.**    Our work examines the computational aspects of Bayesian persuasion, which has gained significant attention since the seminal work of Dughmi and Xu [2016], Dughmi [2017] that study the algorithmic complexity of the sender's optimization tasks in several natural input models. In the multi-receiver settings, computational challenges are more pronounced, as demonstrated by Dughmi [2014], Bhaskar et al. [2016] who showed the hardness of computing the optimal signaling scheme in Bayesian zero-sum games. In addition, Rubinstein [2017] established the hardness of finding even an approximate solution, revealing that obtaining an $\epsilon$-approximate signaling scheme in two-player zero-sum games requires quasi-polynomial time. On the positive side, Babichenko and Barman [2017] studied private persuasion with binary action space and presented an efficient approximation of the optimal signaling scheme. Xu [2020] addressed the tractability of optimizing signaling schemes without externalities between the receiver's payoffs. Recently, Zhou et al. [2022] studied algorithmic information design of public and private signaling in atomic singleton congestion games, achieving efficient exact optimization of optimal signaling schemes when the number of resources is constant.

**Robust Bayesian persuasion.**    Recent research on robust Bayesian persuasion has explored various notions to address the uncertainty and robustness in sender-receiver interactions.. The existing literature can be roughly divided into two main approaches: *max-min models* and *faulty receiver models*.

Maximin models consider scenarios in which the sender aims to maximize their worst-case utility under various types of uncertainty. For instance, Dworczak and Pavan [2022] study a two-layer model where the first layer accounts for arbitrary exogenous information receivers may have, including pessimistic scenarios where receivers fully learn the state from exogenous information. The optimization at the second layer assumes a common prior but is constrained to policies that secure the sender's maximin payoff at the first layer. Kosterina [2022] considers a setting where the sender does know the receiver's prior belief and assumes that the receiver's prior can be any distribution that assigns at least a fixed state-dependent probability to each state. Hu and Weng [2021] consider a model where starting from a common prior, the receiver gets a private signal resulting in an arbitrary random belief supported on a given subset of the simplex. They show (1) when the subset is the entire simplex, full revelation is optimal, and (2) when the subset is a small neighborhood of the common prior, the cost

of robustness vanishes as the size of the neighborhood vanishes. Babichenko et al. [2022] study a model with binary actions and the receiver's utility being uncertain. They show that full uncertainty is extremely harmful, while if the sender knows the ordinal preferences of the receiver over states, then there's a constant additive gap between the maximin strategy and the omniscient one. Chen and Lin [2023] consider a model similar to ours with $\varepsilon$-best responding receivers. They present a continuity result that bounds the cost of robustness by $O(\varepsilon)$.

On the other hand, faulty receiver models usually assume the sender has full knowledge of the receiver's behavioral model, often incorporating bounded rationality or non-standard Bayesian updates. Feng et al. [2024]study the bounded rationality model through quantal response, which is a smoothed version of exact best response that models mistakes or "trembling hands". They focus on binary actions and identify conditions under which the optimal policy against a best-responding receiver remains almost optimal under quantal response. They also construct approximately optimal policies oblivious to the level of bounded rationality. de Clippel and Zhang [2022] consider "non-Bayesian" updates by the receiver according to an arbitrary but fixed rule. They provide structural results and rankings of updating rules in terms of both the sender's and receiver's utilities. These works differ from ours in that they assume the sender has full knowledge about the receiver's fixed response rule, whereas in our setting, the sender must anticipate worst-case responses from a set of approximate best responses.

**Comparison with [Gan et al., 2023].**    Another closely related work is that of Gan et al. [2023] which studies the computational complexity of robust Stackelberg equilibria under approximate best responses. While Stackelberg games provide a more general model for principal-agent interactions, the computational complexity results in [Gan et al., 2023] are not directly applicable to Bayesian persuasion due to differences in problem representation. Specifically, Bayesian persuasion instances are typically represented in a more compact form than the "flat" representation used for Stackelberg games, where principal's strategy space includes all feasible signaling schemes. This distinction limits the direct application of algorithms from Gan et al. [2023] to Bayesian persuasion by framing it as a Stackelberg game instance.

There are also important algorithmic differences between Stackelberg games and Bayesian persuasion. In Stackelberg games, the search space is constrained to the simplex of probability distributions over the principal's action set. In contrast, in Bayesian persuasion, the search space for signaling schemes is not pre-defined as it requires constructing a signal space, making the design of signal spaces an important part of the problem. Our paper introduces algorithmic techniques specifically designed for Bayesian persuasion that achieves a complexity of $\mathrm{poly}(n^{O(m)})$ when the state space size $m$ is small. In comparison, approaches in Gan et al. [2023] result in a complexity exponential in $n$.

# B    Failure of revelation principle

In classical Bayesian persuasion problem where receivers always best respond, an argument via the revelation principle (see, e.g., [Kamenica and Gentzkow, 2011, Proposition 1]) shows that it suffices to consider direct revelation schemes that give direct recommendations of which actions to play. This argument implies that without loss of generality, the number of signals needed is at most the number of actions. However, in this section, we will show that when robustness enters the picture, direct revelation schemes fail even to achieve any nontrivial approximation ratios of the optimal robust utility. Indeed, similar phenomenons were observed by Gan et al. [2023] in the context of Stackelberg games. As they argue, the absence of the revelation principle in their setting makes the respective problems more challenging, which is also what happens in our model.

Below we present a simple example with 3 states and 2 actions where direct revelation schemes fail to match the optimal utility with any nontrivial approximation factor.

**Proposition B.1** (Suboptimality of direct-revelation schemes)**.** *There exists a sequence of Bayesian persuasion instances with a robustness level $\delta = \Theta(1)$, such that the following holds in the limit: any direct-revelation scheme $\varphi$ is suboptimal, at least by a factor of $2$ or an additive gap of $\frac{1}{2}$. That is, for any direct-revelation scheme $\varphi$,*

$$\widehat{S}_\delta(\varphi) \le \frac{1}{2}\widehat{S}_\delta^\star, \quad and \quad \widehat{S}_\delta(\varphi) \le \widehat{S}_\delta^\star - \frac{1}{2}.$$

The claim in Proposition B.1 can be established using the family of robust persuasion instances constructed in Example B.2. The main idea is to create a relatively large robustness level so that disadvantageous actions enter the receiver's approximate best response set as long as the posterior is a mixed distribution. Therefore, to achieve the optimal robust utility, the sender has to adopt a signaling scheme that deterministically discloses the true state, which requires a signaling space larger than the action space.

**Example B.2.** *Consider the following family of robust Bayesian persuasion instances parametrized by $\varepsilon$. Every instance in this family has state space $\Omega = \{\omega_\perp, \omega_0, \omega_1\}$ and action space $\mathcal{A} = \{a_0, a_1\}$. The prior distribution $\mu_0$ puts a very small probability mass on state $\omega_\perp$, and divides the rest of the probability mass evenly between $\omega_1$ and $\omega_2$. Formally, for an infinitesimal $\varepsilon > 0$, we have*

$$\mu_0(\omega_\perp) = \varepsilon, \quad \mu_0(\omega_0) = \mu_0(\omega_1) = \frac{1-\varepsilon}{2}.$$

*The utility functions satisfy $r = s$, where both the receiver obtains utility $1$ if the index of the action matches that of the state (i.e., in the case of $(\omega_1, a_1)$ and $(\omega_0, a_0)$) and $0$ otherwise (see Table 1). The approximate best response level is $\delta = 1$.*

| states ($\Omega$) \ actions ($\mathcal{A}$) | $a_0$ | $a_1$ |
|---|---|---|
| $\omega_\perp$ | 0 | 0 |
| $\omega_0$ | 1 | 0 |
| $\omega_1$ | 0 | 1 |

Table 1: The utility function for both the sender and the receiver. If the state is $\omega_\perp$, then both players have 0 utility. Otherwise, both players get utility 1 if and only if the receiver's action matches the true state.

*Proof of Proposition B.1.* In example B.2, the optimal robust utility $\widehat{S}_\delta^\star$ is at least $1 - \varepsilon$, which can be achieved by a full-revelation signaling scheme $\varphi^\star$ supported on 3 signals. More specifically, $\varphi^\star$ is a deterministic mapping from $\Omega$ to a ternary signal space $\Sigma = \{\sigma_\perp, \sigma_0, \sigma_1\}$, such that $\varphi(\omega_i) \equiv \sigma_i$ for $i \in \{\perp, 0, 1\}$. Since this signaling scheme is deterministic, we have $\varphi^\star(\sigma_i) = \mu_0(\omega_i)$, and each posterior distribution $\mu_{\sigma_i}$ is the degenerate distribution on $\omega_i$. They lead to the following approximate best response sets:

$$\mathsf{BR}_\delta(\mu_{\sigma_\perp}) = \{a_0, a_1\}, \quad \mathsf{BR}_\delta(\mu_{\sigma_0}) = \{a_0\}, \quad \mathsf{BR}_\delta(\mu_{\sigma_1}) = \{a_1\}.$$

Therefore, the sender's robust utility is given by

$$\widehat{S}_\delta(\varphi^\star) = \sum_{\sigma \in \Sigma} \varphi^\star(\sigma) \cdot \min_{a \in \mathsf{BR}_\delta(\mu_\sigma)} s(\mu_\sigma, a)$$

$$= \mu_0(\omega_\perp) \cdot s(\mu_{\sigma_\perp}, a_0) + \mu_0(\omega_0) \cdot s(\mu_{\sigma_0}, a_0) + \mu_0(\omega_1) \cdot s(\mu_{\sigma_1}, a_1)$$

$$= \varepsilon \cdot 0 + \frac{1-\varepsilon}{2} \cdot 1 + \frac{1-\varepsilon}{2} \cdot 1 = 1 - \varepsilon.$$

On the other hand, we claim that any signaling scheme $\varphi : \Omega \to \Sigma$ with $|\Sigma| = 2$ has to suffer a suboptimal utility of $\widehat{S}_\delta(\varphi) \leq \frac{1-\varepsilon}{2}$. To see this, note that if a posterior distribution $\mu_\sigma$ of a signal $\sigma \in \Sigma$ is supported on more than one state, then the receiver's utility of the optimal action is at most $\max\{\mu_\sigma(\omega_0), \mu_\sigma(\omega_1)\} < 1$, which is smaller than the robustness level $\delta = 1$. Therefore, any posterior $\mu_\sigma$ that is a mixed distribution must have $\mathsf{BR}_\delta(\mu_\sigma) = \{a_0, a_1\}$, which implies

$$\min_{a \in \mathsf{BR}_\delta(\mu_\sigma)} s(\mu_\sigma, a) = \min\{s(\mu_\sigma, a_0), s(\mu_\sigma, a_1)\} = \min\{\mu_\sigma(\omega_0), \mu_\sigma(\omega_1)\}.$$

Let us denote the two signals in $\Sigma$ as $\sigma_0$ and $\sigma_1$. At least one of the posterior distributions $\mu_{\sigma_0}$ and $\mu_{\sigma_1}$ has to be supported on $\geq 2$ states because $|\Omega| = 3$. We have two following cases:

- **Case 1. Both signals have mixed posteriors.** In this case, we have

$$\widehat{S}_\delta(\varphi) = \sum_{\sigma \in \Sigma} \varphi(\sigma) \cdot \min_{a \in \mathsf{BR}_\delta(\mu_\sigma)} s(\mu_\sigma, a)$$

$$=\varphi(\sigma_0) \cdot \min\{\mu_{\sigma_0}(\omega_0), \mu_{\sigma_0}(\omega_1)\} + \varphi(\sigma_1) \cdot \min\{\mu_{\sigma_1}(\omega_0), \mu_{\sigma_1}(\omega_1)\}$$
$$\leq \varphi(\sigma_0) \cdot \mu_{\sigma_0}(\omega_0) + \varphi(\sigma_1) \cdot \mu_{\sigma_1}(\omega_0)$$
$$=\mu_0(\omega_0) \cdot \varphi(\omega_0, \sigma_0) + \mu_0(\omega_0) \cdot \varphi(\omega_0, \sigma_1) = \mu_0(\omega_0) = \frac{1-\varepsilon}{2},$$

where the last few steps follow from the Bayes' rule.

- **Case 2. Only one signal has a mixed posterior.** WLOG, assume $\mu_{\sigma_0}$ is a mixed distribution whereas $\mu_{\sigma_1}$ is a pure (degenerate) distribution. It is also WLOG to assume that $\mu_{\sigma_1}$ is a degenerate distribution on $\omega_1$, because $\omega_0$ and $\omega_1$ are symmetric and $\omega_\perp$ has 0 utility under both actions. Therefore, we have that $\sigma_1$ is only sent on state $\omega_1$, and $\varphi(\sigma_1) = \mu_0(\omega_1)$. As for the robust utility, we have

$$\widehat{S}_\delta(\varphi) = \sum_{\sigma \in \Sigma} \varphi(\sigma) \cdot \min_{a \in \mathsf{BR}_\delta(\mu_\sigma)} s(\mu_\sigma, a)$$
$$=\varphi(\sigma_0) \cdot \min\{\mu_{\sigma_0}(\omega_0), \mu_{\sigma_0}(\omega_1)\} + \varphi(\sigma_1) \cdot s(\omega_{\sigma_1}, a_1)$$
$$=\varphi(\sigma_0) \cdot 0 + \varphi(\sigma_1) \cdot 1$$
$$=\mu_0(\omega_1) = \frac{1-\varepsilon}{2},$$

where the second-to-last step follows from $\mu_{\sigma_0}(\omega_1) = 0$ because $\sigma_1$ is only sent on state $\omega_1$.

Combining the two cases above, we arrive at the conclusion that any scheme with two signals has robust utility at most $\frac{1-\varepsilon}{2}$, which is half of the optimal robust utility $\widehat{S}_\delta^\star$. In addition, we note that $\widehat{S}_\delta^\star - \widehat{S}_\delta(\varphi) = \frac{1-\varepsilon}{2}$ and the above analysis holds for any $\varepsilon > 0$. Therefore, sending $\varepsilon \to 0$ proves the $\frac{1}{2}$ additive gap. □

## C Omitted proofs in Section 3

*Proof of Lemma 3.2.* Let $\Sigma' = \{(A, \tilde{a}) \mid A \subseteq \mathcal{A}, \tilde{a} \in A\}$. Given any signaling scheme $\varphi : \Omega \to \Sigma$, we use $\Sigma_{(A,\tilde{a})}$ to denote the subset of signals $\sigma \in \Sigma$ that lead to the same best response $a^\star = \tilde{a}$ and the same $\delta$-BR set $\mathsf{BR}_\delta(\mu_\sigma) = A$, i.e.,

$$\Sigma_{(A,\tilde{a})} = \{\sigma \in \Sigma \mid \sigma \in \Sigma : a^\star(\mu_\sigma) = \tilde{a}, \mathsf{BR}_\delta(\mu_\sigma) = A\}.$$

Consider an alternative signaling scheme $\varphi' : \Omega \to \Sigma'$ that merges signals in each subset $\Sigma_{(A,\tilde{a})}$:

$$\forall (A, \tilde{a}) \in \Sigma', \quad \varphi'(\omega, (A, \tilde{a})) = \sum_{\sigma \in \Sigma_{(A,\tilde{a})}} \varphi(\omega, \sigma).$$

The subsets $\Sigma_{(A,\tilde{a})}$ naturally form a partition of the original signal space $\Sigma$.

We first claim that $a^\star(\mu_{(A,\tilde{a})}) = \tilde{a}, \mathsf{BR}_\delta(\mu_{(A,\tilde{a})}) = A$. The first claim follows directly from the revelation principle — since $\tilde{a}$ is the best response to each $\sigma \in \Sigma_{(A,\tilde{a})}$, it must also be the best response to the merged signal. For the second claim, from the Bayes' rule, the posterior of receiving each signal $(A, \tilde{a})$ under the newly constructed scheme $\varphi'$ satisfies

$$\forall \omega \in \Omega, \quad \mu_{(A,\tilde{a})}(\omega) = \frac{\sum_{\sigma \in \Sigma_{(A,\tilde{a})}} \varphi(\sigma)\mu_\sigma(\omega)}{\sum_{\sigma \in \Sigma_{(A,\tilde{a})}} \varphi(\sigma)}, \tag{5}$$

therefore, for each $a \in A$, we have

$$r(\mu_{(A,\tilde{a})}, a) = \frac{\sum_{\sigma \in \Sigma_{(A,\tilde{a})}} \varphi(\sigma) r(\mu_\sigma, a)}{\sum_{\sigma \in \Sigma_{(A,\tilde{a})}} \varphi(\sigma)} > \frac{\sum_{\sigma \in \Sigma_{(A,\tilde{a})}} \varphi(\sigma)(r(\mu_\sigma, \tilde{a}) - \delta)}{\sum_{\sigma \in \Sigma_{(A,\tilde{a})}} \varphi(\sigma)} = r(\mu_{(A,\tilde{a})}, \tilde{a}) - \delta.$$

Similarly, for each $a \in \mathcal{A} \setminus A$, we can conclude that

$$r(\mu_{(A,\tilde{a})}, a) \leq r(\mu_{(A,\tilde{a})}, \tilde{a}) - \delta,$$

The two cases together established the second claim that $\mathsf{BR}_\delta(\mu_{(A,\tilde{a})}) = A$. Finally, for the sender's utility, we have

$$\varphi'((A,\tilde{a})) \min_{a \in A} s(\mu_{(A,\tilde{a})}, a) = \min_{a \in A} \sum_{\sigma \in \Sigma_{(A,\tilde{a})}} \varphi(\sigma) s(\mu_\sigma, a) \qquad \text{(Bayes' rule in eq. (5))}$$

$$\geq \sum_{\sigma \in \Sigma_{(A,\tilde{a})}} \varphi(\sigma) \min_{a \in A} s(\mu_\sigma, a). \qquad \text{(Jensen's inequality)}$$

Since the above inequalities hold for any $(A, \tilde{a}) \in \Sigma'$, the robust utility satisfies

$$\widehat{S}_\delta(\varphi') = \sum_{(A,\tilde{a}) \in \Sigma'} \varphi'((A,\tilde{a})) \cdot \min_{a \in A} s(\mu_{(A,\tilde{a})}, a)$$

$$\geq \sum_{(A,\tilde{a}) \in \Sigma'} \sum_{\sigma \in \Sigma_{(A,\tilde{a})}} \varphi(\sigma) \cdot \varphi(\sigma) \min_{a \in A} s(\mu_\sigma, a) = \widehat{S}_\delta(\varphi),$$

where the last step follows since $\{\Sigma_{(A,\tilde{a})} \mid (A, \tilde{a}) \in \Sigma'\}$ forms a partition of $\Sigma$.

We have shown that restricting the signal space to $\Sigma' = \{(A, \tilde{a}) \mid A \subseteq \mathcal{A}, \ \tilde{a} \in A\}$ by merging signals never decreases the robust utility. Therefore, there must exist an optimal signaling scheme supported on at most $|\Sigma'| = n \cdot 2^{n-1}$ signals. $\qquad \square$

*Proof of Proposition 3.3.* We use OPT to denote the optimal objective value of Fig. 2, and use $\varphi^\star, x^\star$ to denote the optimal variables that achieve OPT. In addition, let $\varphi^\star : \Omega \to \Sigma$ be a signaling scheme induced by the optimal LP variables $\varphi^\star(\omega, A, \tilde{a})$. Since Fig. 2 is a relaxation of the original problem for computing $\widehat{S}_\delta^\star$, we have $\text{OPT} \geq \widehat{S}_\delta^\star$. Therefore, to establish the optimality of $\varphi^\star$, it suffices to show that $\text{OPT} \leq \widehat{S}_\delta(\varphi^\star)$, because it implies

$$\text{OPT} \leq \widehat{S}_\delta(\varphi^\star) \leq \max_\varphi \widehat{S}_\delta(\varphi) = \widehat{S}_\delta^\star \leq \text{OPT},$$

in which all the inequalities must be tight.

Now we prove $\text{OPT} \leq \widehat{S}_\delta(\varphi^\star)$. Since the third constraint in Fig. 2 ensures that all actions excluded form $A$ must have suboptimality gap $\geq \delta$, $A$ is an overestimate of the true $\delta$-BR set, i.e., $A \supseteq \mathsf{BR}_\delta(\mu_{(A,\tilde{a})})$. Therefore, we have

$$\widehat{S}_\delta(\varphi^\star) = \sum_{(A,\tilde{a}) \in \Sigma} \varphi^\star(A, \tilde{a}) \min_{a \in \mathsf{BR}_\delta(\mu_{(A,\tilde{a})})} s(\mu_{(A,\tilde{a})}, a)$$

$$= \sum_{(A,\tilde{a}) \in \Sigma} \min_{a \in \mathsf{BR}_\delta(\mu_{(A,\tilde{a})})} \sum_{\omega \in \Omega} \mu_0(\omega) \varphi^\star(\omega, A, \tilde{a}) s(\mu_{(A,\tilde{a})}, a)$$

$$\geq \sum_{(A,\tilde{a}) \in \Sigma} \min_{a \in A} \sum_{\omega \in \Omega} \mu_0(\omega) \varphi^\star(\omega, A, \tilde{a}) s(\mu_{(A,\tilde{a})}, a) \qquad (A \supseteq \mathsf{BR}_\delta(\mu_{(A,\tilde{a})}))$$

$$\geq \sum_{(A,\tilde{a}) \in \Sigma} x^\star(A, \tilde{a}) \qquad \text{(first constraint in Fig. 2)}$$

$$= \text{OPT}, \qquad \text{(optimality of } x^\star(A, \tilde{a}) \text{ in Fig. 2)}$$

which completes the proof. $\qquad \square$

## D Omitted Proofs from Section 4

### D.1 Proof of Lemma 4.4

*Proof.* We establish this theorem by showing that any two adjacent polytopes $\Delta_{(A_1,\tilde{a}_1)}, \Delta_{(A_2,\tilde{a}_2)}$ in the $m$-dimensional simplex corresponds to neighboring vertices $(A_1, \tilde{a}_1)$ and $(A_2, \tilde{a}_2)$ in the symmetric difference graph. Since polytopes $\Delta_{(A,\tilde{a})}$ form a partition of the convex simplex $\Delta(\Omega)$, any path through the simplex's polytopes translates into a connected path in the symmetric difference graph $G$, therefore establishing the connectivity of the subgraph induced by $\Sigma^\dagger$.

If $\Delta_{(A_1,\tilde{a}_1)}, \Delta_{(A_2,\tilde{a}_2)}$ are adjacent, they must share a common face, which belongs to a hyperplane in $\mathcal{H}$. Consider the following two cases:

**Case 1.** If the hyperplane is $\sum_{\omega\in\Omega} \boldsymbol{\mu}(\omega)(r(\omega,a) - r(\omega,a')) = 0$ for some pair of actions $(a,a')$. According to the semantics in eq. (3) that defines each polytope, it must be the case that $a, a'$ each corresponds to an action in $\tilde{a}_1, \tilde{a}_2$. WLOG assume $a = \tilde{a}_1$ and $a' = \tilde{a}_2$. Therefore, to show that $\{(A_1, \tilde{a}_1), (A_2, \tilde{a}_2)\} \in E$, it suffices to establish $A_1 = A_2$.

Consider two posteriors $\mu_1 \in \Delta_{(A_1,\tilde{a}_1)}$ and $\mu_2 \in \Delta_{(A_2,\tilde{a}_2)}$ such that $\|\mu_1 - \mu_2\|_1 \leq \varepsilon$ for an infinitesimal $\varepsilon > 0$, and both $\mu_1, \mu_2$ have a distance of at least $\varepsilon_0$ from all other hyperplanes in $\mathcal{H}$. Such a choice implies

- $\forall a \in A,\ |r(\mu_1, a) - r(\mu_2, a)| \leq \|\mu_1 - \mu_2\|_1 = \varepsilon$.

- $r(\mu_1, \tilde{a}_1) - \min_{a\in A_1} r(\mu_1, a) \leq \delta - \varepsilon_0$, and $r(\mu_2, \tilde{a}_2) - \min_{a\in A_2} r(\mu_2, a) \leq \delta - \varepsilon_0$.

Consider the process of fixing $\varepsilon_0$ and sending $\varepsilon \to 0$. For any $a \in A_1 = \mathsf{BR}_\delta(\mu_1)$, we have

$$r(\mu_2, a) \geq r(\mu_1, a) - \varepsilon \geq r(\mu_1, \tilde{a}_1) - (\delta - \varepsilon_0) - \varepsilon \geq r(\mu_2, \tilde{a}_1) - \delta + \varepsilon_0 - 2\varepsilon > r(\mu_2, \tilde{a}_1) - \delta,$$

which implies $a \in \mathsf{BR}_\delta(\mu_2) = A_2$ and therefore $A_1 \subseteq A_2$. In addition, a symmetric argument implies that $A_2 \subseteq A_1$. Therefore, it must be the case that $A_1 = A_2$.

**Case 2.** If the hyperplane is $\sum_{\omega\in\Omega} \boldsymbol{\mu}(\omega)(r(\omega,a) - r(\omega,a')) = \delta$ for some pair of actions $(a,a')$. Once again, we can find two posteriors $\mu_1 \in \Delta_{(A_1,\tilde{a}_1)}$ and $\mu_2 \in \Delta_{(A_2,\tilde{a}_2)}$ such that $\|\mu_1 - \mu_2\|_1 \leq \varepsilon$ for an infinitesimal $\varepsilon > 0$, and both $\mu_1, \mu_2$ have a distance of at least $\varepsilon_0$ from all other hyperplanes in $\mathcal{H}$. This immediately implies $\tilde{a}_1 = \tilde{a}_2$ because otherwise the hyperplane $\sum_{\omega\in\Omega} \boldsymbol{\mu}(\omega)(r(\omega,a_1) - r(\omega,a_2)) = 0$ will have to separate $\mu_1$ and $\mu_2$. A similar continuity argument to that in the previous case shows that $a_1 = a_2 = a$ and $(A_1 \setminus A_2) \cup (A_2 \setminus A_1) = \{a'\}$.

Therefore, in both cases we can conclude that $\{(A_1, \tilde{a}_1), (A_2, \tilde{a}_2)\} \in E$. This finishes the proof of Lemma 4.4. $\qquad\square$

### D.2 Proof of Lemma 4.2

*Proof.* The first bound follows from $|\Sigma^\dagger| \leq |\Sigma| \leq n2^{n-1}$. For the second bound, consider the following set of hyperplanes defined by every pair of receiver's actions:

$$\mathcal{H} = \bigcup_{a,a'\in\mathcal{A}} \left\{ \sum_{\omega\in\Omega} \boldsymbol{\mu}(\omega)(r(\omega,a) - r(\omega,a')) = 0,\ \sum_{\omega\in\Omega} \boldsymbol{\mu}(\omega)(r(\omega,a) - r(\omega,a')) = \delta \right\}$$

They cut the $(m-1)$-dimensional simplex into at most

$$\binom{|\mathcal{H}|}{\leq m} \leq \binom{2n^2}{\leq m} \leq n^{O(m)}$$

cells [Halperin and Sharir, 2017, Theorem 28.1.1]. Since the boundaries of each open polytope $\Delta_{(A,\tilde{a})}$ are defined by a subset of hyperplanes in $\mathcal{H}$ with either strict or non-strict inequalities, each polytopes remain undivided by any hyperplane. Therefore, we have $|\Sigma^\dagger| \leq n^{O(m)}$. $\qquad\square$

### D.3 LP for checking the feasibility of state-action tuples

## E Proof of Theorem 5.1

*Proof.* For any given $\delta \in (0,1)$, we establish the NP-hardness by constructing a reduction from the NP-complete problem of *Subset Sum* (see, e.g., Section 8.8 of [Kleinberg and Tardos, 2006]) to the problem of finding the optimal $\delta$-robust sender utility in Bayesian persuasion against approximate responses. This reduction maps instances of the Subset Sum problem to instances of $\delta$-robust Bayesian persuasion such that an instance of the Subset Sum problem has a solution (a YES instance) if and only if the corresponding $\delta$-robust persuasion instance yields an optimal utility of at least $\frac{1}{2}$.

We start by introducing the format of a subset sum instance. A subset sum instance is given by a set $X = \{x_1, \cdots, x_n\}$ of integers where $\mathsf{sum}(X) = \sum_{i=1}^{n} x_i = 0$. The problem is to decide whether

$$
\begin{aligned}
\underset{\boldsymbol{\varepsilon}, \boldsymbol{\mu}}{\text{maximize}} \quad & \boldsymbol{\varepsilon} \\
\text{subject to} \quad & \sum_{\omega \in \Omega} \boldsymbol{\mu}(\omega)\left(r(\omega, \tilde{a}) - r(\omega, a)\right) \geq 0, && \forall a \in A \\
& \sum_{\omega \in \Omega} \boldsymbol{\mu}(\omega)\left(r(\omega, \tilde{a}) - r(\omega, a)\right) \leq \delta - \boldsymbol{\varepsilon}, && \forall a \in A \\
& \sum_{\omega \in \Omega} \boldsymbol{\mu}(\omega)\left(r(\omega, \tilde{a}) - r(\omega, a')\right) \geq \delta, && \forall a' \in \mathcal{A} \setminus A \\
& \sum_{\omega \in \Omega} \boldsymbol{\mu}(\omega) = 1 \\
& \boldsymbol{\mu}(\omega) \geq 0 && \forall \omega \in \Omega.
\end{aligned}
$$

Figure 4: LP for checking the feasibility of $(A, \tilde{a}) \in \Sigma$. The tuple is feasible if and only if $\varepsilon^\star > 0$.

or not there exists a subset of indices $J \subset [n]$ such that $|J| = \frac{n}{2}$ and $\mathsf{sum}(X_J) = \sum_{j \in J} x_j = 0$. If there exists such a $J$, then $X$ is a YES instance. Otherwise, it is a NO instance. One may check the above version of the problem is also NP-hard. In particular, the requirements that $\mathsf{sum}(X) = 0$ and $|J| = \frac{n}{2}$ are without loss of generality, because one may add an additional integer to offset the sum of all integers if it is nonzero to begin with, as well as enough 0's so there is always a subset of size $\frac{n}{2}$ that sums to 0 iff we have a YES instance. We construct a robust persuasion instance as follows.

**States and prior distribution**   The state space is defined as $\Omega = \{\omega_1, \cdots, \omega_n\}$ with $|\Omega| = |X| = n$, and the prior distribution $\mu_0 = \mathsf{Unif}(\Omega)$ is a uniform distribution over the entire state space.

**Receiver's action space**   The receiver's action space $\mathcal{A} = A \cup B \cup C$ consists of three categories of actions: malicious guessing actions $A = \{a_1, \cdots, a_n\}$, benign guessing actions $B = \{b_1, \cdots, b_n\}$, and sign matching actions $C = \{c_+, c_-\}$.

**Utility functions**   We define the sender's utility $s$ and receiver's utility $r$ as follows. For ease of presentation, the range of the receiver's utility is designed to be $[0, 2]$, but it is not hard to rescale it to $[0, 1]$ to fit our modeling assumptions.

- For a malicious guess $a_i \in A$ and any $i \in [n]$, the utility functions are defined as

$$
r(\omega_j, a_i) = \begin{cases} 1, & i = j \\ \max\left\{1 - \frac{\delta}{1 - \frac{2}{n}}, 0\right\}, & i \neq j \end{cases}, \qquad s(\omega_j, a_i) \equiv 0. \tag{6}
$$

  At a high level, malicious guesses result in a constant $0$ utility on the sender, but the penalty of incorrect guesses on the receiver's side is relatively minor — on top of the base utility of $1$, the receiver only suffers a utility loss of $\frac{\delta}{1 - \frac{2}{n}}$ (when $\delta < 1 - \frac{2}{n}$) for an incorrect state guess, which implies that the receiver secures a utility of $\geq 1 - \delta$ as long as the guess is correct with a probability at least $\frac{2}{n}$. This choice of utility function guarantees that to obtain nontrivial utilities, the sender must prevent the receiver from adopting malicious guesses by creating enough dispersion in the posterior distributions. In particular, the posterior probability on any state should not exceed $\frac{2}{n}$.

- For benign guess $b_i \in B$ for any $i \in [n]$, the utility functions are defined as

$$
r(\omega_j, b_i) = \begin{cases} 1, & i = j \\ 1 - \delta, & i \neq j \end{cases}, \qquad s(\omega_j, b_i) = \begin{cases} 1, & i = j \\ \frac{\frac{1}{2} - \frac{2}{n}}{1 - \frac{2}{n}}, & i \neq j \end{cases}. \tag{7}
$$

  At a high level, benign guesses, if correct, are beneficial for both the sender and receiver. They are intended to encourage the sender to concentrate more probability mass on some certain state in the posterior distribution. To see this, note that as long as the state $\omega_i$ is in the support of the receiver's posterior, the excepted utility of the receiver will exceed $1 - \delta$

and the corresponding benign guessing action $b_i$ enters the $\delta$-BR set. For the sender, the expected utility under malicious guess $b_i$ reaches $\frac{1}{2}$ only when the posterior probability on $\omega_i$ is at least $\frac{2}{n}$. Therefore, to guarantee that the sender has an expected utility of at least $1/2$, the posterior of any state in the support should be no less than $2/n$.

- For the matching actions $C = \{c_+, c_-\}$, the utility functions are defined as

$$r(\omega_j, c_\pm) = 1 \pm \frac{\delta}{4M} x_j, \qquad s(\omega_j, c_\pm) = \frac{1}{2} \mp \frac{1}{4M} x_j, \tag{8}$$

where $M = \sum_{x \in X} |x| > 0$. In this case, the receiver gains utility when her chosen action matches the sign of the expected value of $x_i$ under the posterior distribution, whereas the sender loses a proportional amount of utility. Specifically, when the posterior is uniform on a subspace of size $\frac{n}{2}$ — a design choice that is encouraged by the first two categories of actions — the receiver chooses sign-matching actions based on the sign of the sum of the corresponding subset's elements. In this special case, the sender's utility falls below $< \frac{1}{2}$ unless the subset sum equals zero.

Now, we formally show the reduction from the subset sum problem to $\delta$-robust optimal signaling scheme by dividing the proof into two claims.

**Claim E.1.** *If $X$ is a YES instance, then $\widehat{S}_\delta^\star = \widehat{S}_\delta(\varphi^\star) \geq \frac{1}{2}$.*

*Proof of Claim E.1.* For notation convenience, for any subset of indices $J \subseteq [n]$ and any set $\Omega$ s.t. the elements of $\Omega$ are indexed by $[n]$, we use $X_J$ to denote the subset of elements with indices in $J$, i.e., $X_J = \{\omega_j \mid j \in J\}$. Let $J \subset [n]$ such that $|J| = \frac{n}{2}$ and $\text{sum}(X_J) = \sum_{j \in J} x_j = 0$. Consider the following signaling scheme: $\varphi^\star(\omega_i, \sigma_+) = \mathbb{1}[i \in J]$, $\varphi^\star(\omega_i, \sigma_-) = \mathbb{1}[i \in ([n] \setminus J)]$. Since the signaling scheme is deterministic, we have $\mu_{\sigma_+} = \text{Unif}(\Omega_J)$ and $\mu_{\sigma_-} = \text{Unif}(\Omega_{[n] \setminus J})$.

To compute the robust utility of $\varphi^\star$, we first compute the $\delta$-BR set $\text{BR}_\delta(\mu_{\sigma_+})$. For the sign-matching actions $c \in C$, we have

$$r(\mu_{\sigma_+}, c) = 1 \pm \frac{\delta}{4M} \cdot \frac{2}{n} \sum_{j \in J} x_j = 1.$$

because the subset $X_J$ is of sum $0$. For malicious guesses $a_i \in A$, note that $\mu_{\sigma_+}(\omega_j) \leq 2/n$ for any $\omega_j \in \Omega$, thus we have

$$r(\mu_{\sigma_+}, a_i) \leq \frac{2}{n} + \left(1 - \frac{2}{n}\right) \cdot \left(1 - \frac{\delta}{1 - \frac{2}{n}}\right) = 1 - \delta.$$

For any benign guesses $b_i \in B$, we have

$$r(\mu_{\sigma_+}, b_i) = 1 \cdot \mu_{\sigma_+}(\omega_i) + (1 - \delta)(1 - \mu_{\sigma_+}(\omega_i)) = 1 - \delta(1 - \mu_{\sigma_+}(\omega_i)),$$

which implies that $r(\mu_{\sigma_+}, b_i) > 1 - \delta$ if and only if $\mu_{\sigma_+}(\omega_i) > 0$. Therefore, we have $\text{BR}_\delta(\mu_{\sigma_+}) = B_J \cup C$.

As for the sender's robust utility, we have

$$s(\mu_{\sigma_+}, b_i) = 1 \cdot \frac{2}{n} + \left(1 - \frac{2}{n}\right) \cdot \frac{\frac{1}{2} - \frac{2}{n}}{1 - \frac{2}{n}} = \frac{1}{2}, \qquad\qquad \forall b_i \in B_J;$$

$$s(\mu_{\sigma_+}, c) = \frac{1}{2} \mp \frac{\delta}{4M} \cdot \frac{2}{n} \sum_{j \in J} x_j = \frac{1}{2} \mp \frac{\delta}{4M} \cdot \frac{2}{n} \text{sum}(X_J) = \frac{1}{2}, \qquad \forall c \in C.$$

Therefore, we have $\min_{a \in \text{BR}_\delta(\sigma_+)} s(\mu_{\sigma_+}, a) = \frac{1}{2}$.

A similar analysis show that $\text{BR}_\delta(\mu_{\sigma_-}) = B_{[n] \setminus J} \cup C$ and $\min_{a \in \text{BR}_\delta(\sigma_-)} s(\mu_{\sigma_-}, a) = \frac{1}{2}$. Therefore,

$$\widehat{S}_\delta^\star \geq \widehat{S}_\delta(\varphi^\star) = \varphi^\star(\sigma_+) \min_{a \in \text{BR}_\delta(\sigma_+)} s(\mu_{\sigma_+}, a) + \varphi^\star(\sigma_-) \min_{a \in \text{BR}_\delta(\sigma_-)} s(\mu_{\sigma_-}, a) = \frac{1}{2}. \qquad \square$$

**Claim E.2.** *If the optimal sender's utility satisfies $\widehat{S}_\delta^\star \geq \frac{1}{2}$, then $X$ must be a YES instance.*

*Proof of Claim E.2.* For every posterior distribution $\mu_\sigma$, we show that the worst-case expected sender utitiliy $f_\delta(\mu_\sigma) \triangleq \min_{a \in \mathsf{BR}_\delta(\mu_\sigma)} s(\mu_\sigma, a) \leq \frac{1}{2}$, and the equality holds only when $\mu_\sigma = \mathsf{Unif}(X_J)$ where $|J| = \frac{n}{2}$ and $\mathsf{sum}(X_J) = 0$. Therefore, if $\widehat{S}^\star_\delta \geq \frac{1}{2}$, then $f_\delta(\mu_\sigma) \leq \frac{1}{2}$ must hold with equality for all $\sigma \in \Sigma$, which implies the existence of a subset with size $\frac{n}{2}$ and sum 0, i.e., $X$ is a YES instance.

Consider the following cases:

**Case 1.** If $a_i \in \mathsf{BR}_\delta(\mu_\sigma) \cap A$ for some $i \in [n]$, then $f_\delta(\mu_\sigma) \leq s(\mu_\sigma, a_i) = 0 < \frac{1}{2}$.

**Case 2.** IF $\mathsf{BR}_\delta(\mu_\sigma) \cap A = \emptyset$. Consider two subcases based on whether or not $r(\mu_\sigma, a^\star(\mu_\sigma)) > 1$. In the case of $r(\mu_\sigma, a^\star(\mu_\sigma)) > 1$, it must be that $a^\star(\mu_\sigma) \in C$ because other actions all have receiver utility $\leq 1$. From the design of sign-matching utilities, the sender's utility exceeds $\frac{1}{2}$ if and only if the receiver's utility is strictly lower than 1, so we must have $f_\delta(\mu_\sigma) \leq s(\mu_\sigma, a^\star(\mu_\sigma)) < \frac{1}{2}$. Therefore, it remains to consider the case of $r(\mu_\sigma, a^\star(\mu_\sigma)) \leq 1$.

On the one hand, combining with the fact that $\mathsf{BR}_\delta(\mu_\sigma) \cap A = \emptyset$, this gives

$$\max_{a_i \in A} r(\mu_\sigma, a_i) \leq r(\mu_\sigma, a^\star(\mu_\sigma)) - \delta \leq 1 - \delta \implies \max_{i \in [n]} \mu_\sigma(\omega_i) \leq \frac{2}{n}, \tag{9}$$

Which implies that all posterior probabilities should be no larger than $\frac{2}{n}$. On the other hand, let $J = \mathsf{supp}(\mu_\sigma)$, then for all $i \in J$, we have

$$r(\mu_\sigma, b_i) > 1 - \delta \geq r(\mu_\sigma, a^\star(\mu_\sigma)) - \delta,$$

which implies $B_J \subseteq \mathsf{BR}_\delta(\mu_\sigma)$. We further consider the following two subcases:

- If $\exists i \in J$, s.t. $\mu_\sigma(\omega_i) < \frac{2}{n}$, then we have $f_\delta(\mu_\sigma) \leq s(\mu_\sigma, b_i) < 1 \cdot \frac{2}{n} + \left(1 - \frac{2}{n}\right) \cdot \frac{\frac{1}{2} - \frac{2}{n}}{1 - \frac{2}{n}} = \frac{1}{2}$.

- Otherwise, we have $\min_{j \in J} \mu_\sigma(\omega_j) \geq \frac{2}{n}$, i.e., all non-zero posterior probabilities should be no smaller than $\frac{2}{n}$. Since Equation (9) implies that all posterior probabilities should also be no larger than $\frac{2}{n}$, it must be the case that they are exactly $\frac{2}{n}$, i.e., $\mu_\sigma$ is a uniform distribution supported on exactly $|J| = \frac{n}{2}$ elements. Therefore,

$$f_\delta(\mu_\sigma) \leq \min_{i \in J} s(\mu_\sigma, b_i) = 1 \cdot \frac{2}{n} + (1 - 2/n) \cdot \frac{\frac{1}{2} - \frac{2}{n}}{1 - \frac{2}{n}} = \frac{1}{2}.$$

  Moreover, the uniform distribution also implies $r(\mu_\sigma, c_\pm) = 1 + \frac{\delta}{4M} \cdot \frac{2}{n} \cdot \left(\mathsf{sum}(X_J)\right)_\pm$. Together with the fact that $r(\mu_\sigma, a^\star(\mu_\sigma)) \leq 1$, we have

$$r(\mu_\sigma, c_{\mathsf{sgn}(\mathsf{sum}(X_J))}) = 1 + \frac{\delta}{2Mn}|\mathsf{sum}(X_J)| \leq r(\mu_\sigma, a^\star(\mu_\sigma)) \leq 1 \implies \mathsf{sum}(X_J) = 0. \tag{10}$$

  Therefore, $J \subset [n]$ is a subset of size $\frac{n}{2}$ that satisfies $\mathsf{sum}(X_J) = 0$, and thus $X$ is a YES instance. In this case, we have $\mathsf{BR}(\mu_\sigma) = B_J \cup C$ and $f_\delta(\mu_\sigma) = \frac{1}{2}$.

In conclusion, we have shown that all but the last case have $f_\delta(\mu_\sigma) < \frac{1}{2}$, whereas the last case has both $f_\delta(\mu_\sigma) = \frac{1}{2}$ and that the subset sum problem is a YES instance. This finishes the proof of the claim. $\square$

Combining Claim E.1 and Claim E.2 implies that $\widehat{S}^\star_\delta \geq \frac{1}{2}$ if and only if the subset sum problem $X$ is a YES instance. We have thus established a reduction from the subset sum problem to that of computing the optimal $\delta$-robust signaling scheme, which proves the NP-hardness of the later problem. $\square$

# F  Proof of Theorem 5.2

*Proof of Theorem 5.2.* Let $\mathsf{OPT}$ be the optimal objective value and let $\widehat{\varphi}$ be the signaling scheme induced by the optimal solution $\varphi^\star$. We divide the proof into two parts: the first part proves

$\mathsf{OPT} \geq \widehat{S}_\delta^\star - \varepsilon'$, and the second part proves $\widehat{S}_\delta(\widehat{\varphi}) \geq \mathsf{OPT} - 4\varepsilon'$. Putting both together establishes the claim because

$$\widehat{S}_\delta(\widehat{\varphi}) \geq \mathsf{OPT} - 4\varepsilon' \geq \widehat{S}_\delta^\star - 4\varepsilon' - \varepsilon' = \widehat{S}_\delta^\star - 5\varepsilon' = \widehat{S}_\delta^\star - \varepsilon,$$

where the last step follows from setting $\varepsilon' = \frac{\varepsilon}{5}$ in Fig. 3.

To prove the first inequality of $\mathsf{OPT} \geq \widehat{S}_\delta^\star - \varepsilon'$, we show that the optimal robust signaling scheme can be discretized to fit in our LP with no more than $\varepsilon'$ discretization error. Specifically, let $\varphi : \Omega \to \Sigma$ be an optimal signaling scheme that has $\widehat{S}(\varphi_1^\star) = \widehat{S}_\delta^\star$. We will construct LP variables $\varphi$ that are feasible in Fig. 3.

For each $\sigma \in \Sigma$. there exists $\overline{\mu}_\sigma \in \mathcal{G}_k$ such that $\forall a \in \mathcal{A}$, $|s(\mu_\sigma, a) - s(\overline{\mu}_\sigma, a)| \leq \varepsilon$. let $\Sigma_{(\overline{\mu}, \tilde{a}, \dot{a})} \subseteq \Sigma$ be the subset of signals $\sigma \in \Sigma$ with $\overline{\mu} = \overline{\mu}_\sigma$, $a^\star(\mu_\sigma) = \tilde{a}$, and

$$\dot{a} \in \operatorname*{argmin}_{a : r(\mu_\sigma, a) > r(\mu_\sigma, \tilde{a}) - \delta} s(\overline{\mu}_\sigma, a) \tag{11}$$

We define $\varphi$ as merging the signals in $\Sigma_{(\overline{\mu}, \tilde{a}, \dot{a})}$. Formally,

$$\varphi(\omega, \overline{\mu}, \tilde{a}, \dot{a}) \triangleq \sum_{\sigma \in \Sigma_{(\overline{\mu}, \tilde{a}, \dot{a})}} \varphi(\omega, \sigma). \tag{12}$$

Such a construction immediately satisfy the last two constraints the LP, because all the subsets $\Sigma_{(\overline{\mu}, \tilde{a}, \dot{a})} \subseteq \Sigma$ where $(\overline{\mu}, \tilde{a}, \dot{a}) \in \widehat{\Sigma}$ form a partition of the original signal space $\Sigma$. The first two constraints are also satisfied because for each $\overline{\mu} \in \mathcal{G}_k$,

$$\sum_{\omega \in \Omega} \mu_0(\omega) \varphi(\omega, \overline{\mu}, \tilde{a}, \dot{a}) \cdot (s(\omega, a') - s(\overline{\mu}, a) + \varepsilon)$$

$$= \sum_{\omega \in \Omega} \mu_0(\omega) \sum_{\sigma \in \Sigma_{(\overline{\mu}, \tilde{a}, \dot{a})}} \varphi(\omega, \sigma) \cdot (s(\omega, a') - s(\overline{\mu}_\sigma, a) + \varepsilon) \qquad \text{(definition of } \varphi \text{ in eq. (12))}$$

$$\geq \sum_{\omega \in \Omega} \mu_0(\omega) \sum_{\sigma \in \Sigma_{(\overline{\mu}, \tilde{a}, \dot{a})}} \varphi(\omega, \sigma) \cdot (s(\omega, a') - s(\mu_\sigma, a)) \qquad (|s(\mu_\sigma, a) - s(\overline{\mu}_\sigma, a)| \leq \varepsilon)$$

$$= \sum_{\sigma \in \Sigma_{(\overline{\mu}, \tilde{a}, \dot{a})}} \sum_{\omega \in \Omega} \mu_0(\omega) \varphi(\omega, \sigma) \left( s(\omega, a') - s(\mu_\sigma, a) \right) = 0, \qquad \text{(Bayes' rule)}$$

and a similar analysis verifies $\sum_{\omega \in \Omega} \mu_0(\omega) \varphi(\omega, \overline{\mu}, \tilde{a}, \dot{a}) \cdot (s(\omega, a') - s(\overline{\mu}, a) - \varepsilon) \leq 0$. For the third constraint, since $\tilde{a}$ is the best response to every $\sigma \in \Sigma_{(\overline{\mu}, \tilde{a}, \dot{a})}$, it should also be the best response to the combined signal $(\overline{\mu}, \tilde{a}, \dot{a})$. For the fourth constraint, for a tuple $(\overline{\mu}, \tilde{a}, \dot{a}) \in \widehat{\Sigma}$ and an action $a$ such that $s(\overline{\mu}, a) < s(\overline{\mu}, \dot{a}) - 2\varepsilon'$, we have

$$\forall \sigma \in \Sigma_{(\overline{\mu}, \tilde{a}, \dot{a})}, \quad s(\mu_\sigma, a) \leq s(\overline{\mu}, a) + \varepsilon' < s(\overline{\mu}, \dot{a}) - \varepsilon' \leq s(\mu_\sigma, \dot{a})$$

Together with the choice of $\dot{a}$ in Equation (11) that $\dot{a}$ minimizes $s(\mu_\sigma, a)$ for all $a \in \mathsf{BR}_\delta(\mu_\sigma)$, we have $r(\mu_\sigma, a) \leq r(\mu_\sigma, \dot{a})$ for every $\sigma \in \Sigma_{(\overline{\mu}, \tilde{a}, \dot{a})}$. Therefore, the fourth constraint is satisfied as

$$\sum_{\omega \in \Omega} \mu_0(\omega) \varphi(\omega, \overline{\mu}, \tilde{a}, \dot{a}) \left( r(\omega, \tilde{a}) - r(\omega, a) - \delta \right) = \sum_{\sigma \in \Sigma_{(\overline{\mu}, \tilde{a}, \dot{a})}} \varphi(\sigma) \left( r(\mu_\sigma, \tilde{a}) - r(\mu_\sigma, a) - \delta \right) \geq 0.$$

Finally, for the LP's objective value, we have

$$\mathsf{OPT} \geq \sum_{\omega \in \Omega} \mu_0(\omega) \sum_{(\overline{\mu}, \tilde{a}, \dot{a}) \in \widehat{\Sigma}} \varphi(\omega, \overline{\mu}, \tilde{a}, \dot{a}) s(\omega, \dot{a})$$

$$\geq \sum_{\omega \in \Omega} \mu_0(\omega) \sum_{(\overline{\mu}, \tilde{a}, \dot{a}) \in \widehat{\Sigma}} \varphi(\omega, \overline{\mu}, \tilde{a}, \dot{a}) \left( s(\overline{\mu}, \dot{a}) - \varepsilon' \right) \qquad \text{(the first constraint)}$$

$$\geq \sum_{\omega \in \Omega} \mu_0(\omega) \sum_{\sigma \in \Sigma} \varphi(\omega, \sigma) \left( s(\overline{\mu}_\sigma, \dot{a}) - \varepsilon' \right) \qquad (\overline{\mu}_\sigma = \overline{\mu} \text{ for all } \sigma \in \Sigma_{(\overline{\mu}, \tilde{a}, \dot{a})})$$

$$= \sum_{\sigma \in \Sigma} \min_{a \in \mathsf{BR}_\delta(\mu_\sigma)} \sum_{\omega \in \Omega} \mu_0(\omega) \varphi(\omega, \sigma) s(\overline{\mu}_\sigma, a) - \varepsilon' \qquad \text{(definition of } \dot{a} \text{ in eq. (11))}$$

$$=\widehat{S}(\varphi^\star) - \varepsilon' = \widehat{S}_\delta^\star - \varepsilon'.$$

We proceed to prove the second part of the theorem, namely the signaling scheme $\widehat{\varphi}$ induced by optimal LP solution $\varphi^\star$ satisfies $\widehat{S}_\delta(\widehat{\varphi}) \geq \mathsf{OPT} - 4\varepsilon'$. The key step to establishing this claim is to show that, $\forall(\overline{\mu}, \tilde{a}, \dot{a}) \in \widehat{\Sigma}$, the sender's utility under the worst-case $\delta$-BR strategy $\rho_\delta((\overline{\mu}, \tilde{a}, \dot{a}))$ is not much worse than that under action $\dot{a}$. We establish this using proof by contradiction. Suppose an action $a = \rho_\delta((\overline{\mu}, \tilde{a}, \dot{a})) \in \mathsf{BR}_\delta(\mu_{\overline{\mu}, \tilde{a}, \dot{a}})$ satisfies $s(\mu_{(\overline{\mu}, \tilde{a}, \dot{a})}, a) < s(\mu_{(\overline{\mu}, \tilde{a}, \dot{a})}, \dot{a}) - 4\varepsilon'$, then we should have $s(\overline{\mu}, a) < s(\overline{\mu}, a) - 2\varepsilon'$ because the first and second constraints of the LP in Fig. 3 ensures

$$s(\overline{\mu}, a) \leq s(\mu_{(\overline{\mu}, \tilde{a}, \dot{a})}, a) + \varepsilon' < s(\mu_{(\overline{\mu}, \tilde{a}, \dot{a})}, \dot{a}) - 3\varepsilon' \leq s(\overline{\mu}, \dot{a}) - 2\varepsilon'.$$

However, from the fourth constraint, in the case of $s(\overline{\mu}, a) < s(\overline{\mu}, a) - 2\varepsilon'$, we have

$$\sum_{\omega \in \Omega} \mu_0(\omega)\varphi(\omega, \overline{\mu}, \tilde{a}, \dot{a})\left(r(\omega, \tilde{a}) - r(\omega, a) - \delta\right) \geq 0$$
$$\Longleftrightarrow r(\mu_{(\overline{\mu}, \tilde{a}, \dot{a})}, a) \leq r(\mu_{(\overline{\mu}, \tilde{a}, \dot{a})}, \dot{a}) - \delta,$$

which contradicts the assumption of $a \in \mathsf{BR}_\delta(\mu_{(\overline{\mu}, \tilde{a}, \dot{a})})$. Therefore, we must have

$$s(\mu_{(\overline{\mu}, \tilde{a}, \dot{a})}, a) \geq s(\mu_{(\overline{\mu}, \tilde{a}, \dot{a})}, \dot{a}) - 4\varepsilon'. \tag{13}$$

Finally, for the sender's robust utility, we have

$$\widehat{S}_\delta(\widehat{\varphi}) = \sum_{\omega \in \Omega} \mu_0(\omega) \sum_{(\overline{\mu}, \tilde{a}, \dot{a}) \in \widehat{\Sigma}} \varphi(\omega, \overline{\mu}, \tilde{a}, \dot{a}) \cdot s(\omega, \rho_\delta((\overline{\mu}, \tilde{a}, \dot{a})))$$

$$= \sum_{(\overline{\mu}, \tilde{a}, \dot{a}) \in \widehat{\Sigma}} \widehat{\varphi}((\overline{\mu}, \tilde{a}, \dot{a})) \cdot s(\mu_{(\overline{\mu}, \tilde{a}, \dot{a})}, \rho_\delta((\overline{\mu}, \tilde{a}, \dot{a}))) \qquad \text{(Bayes' rule)}$$

$$\geq \sum_{(\overline{\mu}, \tilde{a}, \dot{a}) \in \widehat{\Sigma}} \widehat{\varphi}((\overline{\mu}, \tilde{a}, \dot{a})) \cdot \left(s(\mu_{(\overline{\mu}, \tilde{a}, \dot{a})}, \dot{a}) - 4\varepsilon'\right) \qquad \text{(From Equation (13))}$$

$$= \sum_{\omega \in \Omega} \mu_0(\omega) \sum_{(\overline{\mu}, \tilde{a}, \dot{a}) \in \widehat{\Sigma}} \varphi(\omega, \overline{\mu}, \tilde{a}, \dot{a}) \cdot s(\omega, \dot{a}) - 4\varepsilon' \qquad \text{(Bayes' rule)}$$

$$= \mathsf{OPT} - 4\varepsilon'. \qquad\qquad \square$$

