# OpenReview forum: "Computational Aspects of Bayesian Persuasion under Approximate Best Response"
_NeurIPS.cc/2024/Conference — NeurIPS 2024 poster_

### Official Review · Reviewer_zPak · 2024-07-12

**Soundness:** 3
**Presentation:** 3
**Contribution:** 2
**Rating:** 5
**Confidence:** 3

**Summary:**

The paper considers the problem of BP under \delta-best responses of the receivers. This means that there might be multiple actions that are BR to a specific signalling scheme. This creates non trivial problems of the algorithmic problem of computing the optimal signalling scheme. The paper provides poly-time algorithms with constant number of actions or states and quasi-polynomial algorithms for general case.

**Strengths:**

The paper extends the problem of stackelberg equilibria with delta-BR to the special case of Bayesian persuasion. This is an interesting problem and has somewhat different flavour then the general stakelberg.

**Weaknesses:**

I would like to know more about the connection between the robust Stackelberg paper. Why do you need to prove again the hardness and do not reduct from the hardness of the robust Stackelberg paper? What are the differences between the results there and here. The current work for sure cite that one but fails to discuss properly what is implied and what is not. I feel that I need to be convinced that even if the results and the techniques of the two papers are similar, this one deserves a spot at neurips. I hope that the authors would discuss this in details in the rebuttal and then add such the discussion to the new version of the paper.

**Questions:**

see weaknesses

**Limitations:**

yes

---

> ### Author Rebuttal · Authors · 2024-08-07
>
> Thank you for your thoughtful comments.
>
> **Need for a new reduction**: the fundamental reason that the two problems are not necessarily "comparably hard" is that the computational complexity depends crucially on (1) the representation and (2) additional special structure.  It is certainly true that the Bayesian persuasion problem can be viewed as a Stackelberg game, but:
> - The flat representation of the Stackelberg game corresponding to a Bayesian persuasion instance has a huge strategy space.  In particular, each candidate strategy of the sender is a (randomized) mapping from states to signals.  This means the sender's strategy space is at least exponentially larger even in the classical setting where the revelation principle holds.  As a result, a polynomial-time algorithm under the Stackelberg game representation might very well be an exponential-time algorithm under the Bayesian persuasion representation.  This in particular means the Bayesian persuasion version might not be easier than the Stackelberg game version.
> - On the other hand, Bayesian persuasion is a *special* class of Stackelberg games, which means it might exhibit additional structure that makes the computational problem much easier despite the difference in representations.  This in particular means the Bayesian persuasion version might not be harder than the Stackelberg game version.  Indeed, our proof of hardness directly reduces from Subset Sum.
>
> **Comparison with Gan et al.**: below we provide a detailed comparison to the work by Gan et al.  We will focus on concrete differences, which are easier to describe and verify.  We will also include a shortened version in the main paper (if there's space), as well as the full comparison in an appendix.
>
> - Model: allowing the "agent" / "follower" / "receiver" to choose a response that is suboptimal by a given amount is a standard approach in algorithmic game theory when robustness is desired.  Both our work and that of Gan et al. take this approach.  However, the first major difference (both conceptual and technical) already shows in the respective models: the succinct representation of a Bayesian persuasion instance has the additional component of *states*.  This in particular means the sender's strategy is a randomized mapping from states to posterior beliefs, which is, superficially speaking, of much higher dimensions than a Stackelberg equilibrium.  In fact, since the classical revelation principle is no longer valid, one may suspect the former strategy space is infinite-dimensional (we show this is not the case).  As we argue below, this has significant technical implications to the computation of an (almost) optimal strategy.
>
> - Structure of optimal strategies: our positive results rely crucially on a structural property of optimal strategies that we prove (Lemma 3.2), which doesn't have a counterpart in robust Stackelberg games.  The property says that while there are infinitely many possible signals, restricted to optimal strategies, many of them can be grouped, and we only need to consider a finite number of representative signals.  This in particular means the effective strategy space of the sender is finite-dimensional.  Note that here, the receiver may choose different actions depending on the signal sent.  In contrast, the leader's strategy space in a robust Stackelberg game is naturally finite dimensional, and the follower always chooses a fixed action in response to the leader's strategy.
>
> - Algorithm for fixed number of actions: our algorithm when the number of actions is fixed (Proposition 3.3) is a natural combination of the classical LP for Bayesian persuasion and the above structural property.  In particular, we solve a single LP for the sender's optimal strategy.  For comparison, the fixed-$n$ algorithm by Gan et al. generalizes the algorithm by Conitzer and Sandholm ("Computing the Optimal Strategy to Commit to"), which enumerates the follower's response and solves one LP for each possibility.
>
> - Algorithm for fixed number of states: here we deviate significantly from existing techniques.  In particular, our algorithm relies crucially on the notion of symmetric difference graphs and connectivity therein.  To our knowledge, such techniques have not been employed in the context of Bayesian persuasion or Stackelberg games.  In contrast, there are no states in Stackelberg games to begin with.  Note that the parameter $m$ in Stackelberg games plays an intrinsically different role than the number of states in our model, and Gan et al. present no efficient algorithm when $m$ is fixed (though the comparison itself may not be meaningful in the first place).
>
> - Hardness result: first we note that it's not uncommon for a well-motivated problem to be computationally hard, and the fact that both our problem and Gan et al.'s are hard doesn't necessarily mean the two are otherwise similar (also see our response to the first question above).  Our hardness result is based on a fundamentally different reduction from Gan et al.'s.  In particular, we reduce from the problem of Subset Sum, whereas Gan et al. reduce from Exact Cover by 3-Sets.  Details of the two reductions bear virtually no similarity.
>
> - Approximation algorithm: our algorithm shares the same high-level idea with Gan et al.'s (as well as many other approximation algorithms involving the probability simplex): one first discretizes the probability simplex into a reasonable number of representative points, and then considers the problem restricted to these points.  Despite this high-level similarity, the concrete algorithms are sufficiently different.  Specifically, the two algorithms are based on their exact (and inefficient) versions respectively, which means our algorithm solves a single LP, and Gan et al.'s enumerates the follower's response and solves one LP for each possibility.  Our algorithm also has to deal with additional challenges introduced by the states and the prior distribution.

---

> > ### Comment · Reviewer_zPak · 2024-08-12
> >
> > I'm not convinced by the authors response about the intrinsic differences between their work and the robust Stakelberg paper. For example the fact that you have to deal in principle with infinite strategies is obviously also true for Stakelberg games, and also in Stakelberg games many of those can be grouped (as your second bullet point).
> > However I agree that the poly-time algorithms with constant number of actions or states are interesting and should be given more space.
> > That said I cannot increase my score that was already somewhat on the positive side

---

> > > ### Author Response · Authors · 2024-08-14
> > >
> > > Thank you for your response. We would like to highlight the difference in the search space between the Stackelberg game and the Bayesian persuasion problem (although both are infinite). In Stackelberg games, the search space is constrained to the simplex of probability distributions over the principal's action set. In contrast, in Bayesian persuasion, the search space of signaling schemes is not pre-defined because one needs to define the signal space first, where designing signal space is also an important part of the problem. We also appreciate the reviewer's acknowledgment of our result with small state/action spaces, and we will include a more detailed discussion in revised versions of the paper.

---

### Official Review · Reviewer_R1Q8 · 2024-07-13

**Soundness:** 3
**Presentation:** 4
**Contribution:** 3
**Rating:** 7
**Confidence:** 4

**Summary:**

This paper studies a variant of the Bayesian persuasion problem where, instead of best-responding, the receiver $\delta$-approximately best responds to the sender. Specifically, upon receiving a signal, the receiver takes the $\delta$-optimal action that is worst for the sender on the induced posterior belief. The authors study the complexity of computing the optimal robust signaling scheme for the sender, obtaining four results:

(1) Unlike the classical best-response model where direct-revelation signaling schemes are sufficient to be optimal, direct-revelation schemes are sub-optimal by a factor of 2 in the approximate-best-response model.

(2) The authors give a linear program with size $O(2^n nm)$ to compute an optimal robust signaling scheme (which is not a direct-revelation scheme); $n$ is the number of actions and $m$ is the number of states. The signal in this signaling scheme is interpreted as a tuple $\sigma = (A, \tilde a)$ of a set of $\delta$-best-responding actions $A$ and a best-responding action $\tilde a$.

(3) Then, based on the observation that the number of feasible tuples $\sigma = (A, \tilde a)$ cannot be more than $O(n^{O(m)})$, the authors design an algorithm to compute the optimal robust signaling scheme with $poly( n^{O(m)} )$ complexity.

(4) Finally, for the case of large $n$ and $m$, the authors show NP-hardness of exactly computing the optimal robust signaling scheme and give a quasi-polynomial-time approximation algorithm.

**Strengths:**

(1) [Significance] Bayesian persuasion with an approximately-best-responding receiver is a natural extension to the classical best-response model. As showed by the authors, this problem presents significant technical challenges because the classic idea of restricting to direct-revelation schemes no longer works. So, this problem is both conceptually and technically interesting.

(2) [Quality] The results are comprehensive and non-trivial. Both positive and negative results for the general case are given. And positive results for the special cases of constant number of actions and constant number of states are also given.

(3) [Originality] To design an efficient algorithm for the case of constant number of states (Section 4), the authors make the key observation that the number of feasible tuples is $O(n^{O(m)})$, proved using a fundamental theorem in computational geometry. This observation is unexpected at first sight and the connection with computational geometry is interesting.

(4) [Quality] Discussion of related works, especially Appendix A, is comprehensive and clear.

(5) [Clarity] Writing is clear. The introduction nicely summarizes the technical contributions and high-level ideas.

**Weaknesses:**

I don't see significant weaknesses.

I only have a minor concern regarding the fit to NeurIPS. Most of the algorithmic game theory papers on NeurIPS are related to machine learning in some ways, but this paper is a pure AGT paper with no obvious machine learning components (at least in my opinion).

**Questions:**

(Q1) What is the running time of the QPTAS you design for the general case (Theorem F.1)?

**Limitations:**

**Suggestions:**

(1) Typo: The equation between Lines 192 and 193: "$r(\mu_\sigma, a)$".

(2) Typo: the $s$ in Definition 2.1 should be $\sigma$.

(3) Typo: In Remark 2.2, the phrase "that achieves sender's objective" is confusing and can be deleted.

(4) It is better to explicitly write $\sigma = (A, \tilde a)$ in the first contraint of the optimization problem in the Figure 1.

---

> ### Author Rebuttal · Authors · 2024-08-07
>
> Thank you for your detailed and insightful comments.
>
> **Suggestions**: Thank you for the helpful suggestions, we will revise our paper accordingly.
>
> **Running time of the QPTAS**: the running time generally depends on the specific algorithm chosen to solve the LP given in Theorem F.1.  Algorithms for LP are quickly evolving, but we can apply their result as a black box to obtain running time guarantees in our problem.  From a quick search, the state-of-the-art algorithm (or something close enough) for general LP seems to run in time $\tilde{O}(n^{2 + 1/18}L)$, where $n$ is the total number of variables and constraints, and $L$ is the number of bits required to encode input numbers.  Combining this with the bound on the size of the LP in Theorem F.1, we get a running time of $\tilde{O}(n^4 m^{25\log(2n) / \varepsilon^2} L)$.  It might be possible to apply special-purpose LP algorithms that exploit the structure of our LP formulation to get a better running time.  In any case, the running time is polynomial in $n$, $m^{\log n / \varepsilon}$, and $L$ (the number of bits required to encode input numbers).

---

> > ### Comment · Reviewer_R1Q8 · 2024-08-13
> > **I think this paper has enough technique contribution to the literature**
> >
> > I am happy with the authors' response and keep my rating of 7.  Below, I'd like to discuss my support for this paper.
> >
> > I read other reviews.  A common issue raised by other reviewers is the similarity to [Gan et al (2023), Robust Stackelberg Equilibria].  I am convinced by the authors' response to reviewer zPak that their work is significantly different from [Gan et al (2023)].  Although Bayesian persuasion and Stackelberg games are similar at a high level (they both belong to a class of Generalized Principal-Agent Problem as pointed out by reviewer FR8F), they require different techniques.  For example, in Stackelberg games, the leader only needs to choose one mixed strategy (and this mixed strategy is unconstrained), while in Bayesian persuasion, the sender needs to choose a distribution over posteriors (subject to the constraint that the average of posteriors is equal to the prior) and the support of this distribution might be infinite.  These two scenarios are significantly different just as unconstrained and constrained convex optimization problems are significantly different. The authors also mentioned other differences with [Gan et al, 2023] in their rebuttal.  Although some of the authors' techniques for robust Bayesian persuasion are inspired by [Gan et al (2023)]'s techniques for robust Stackelberg games, the latter does not apply directly to Bayesian persuasion, as the authors argue in their rebuttal.  So, I think this work has enough technical contribution to the literature, although the authors didn't discuss their contributions clearly in their submitted draft.
> >
> > Another contribution of this work that was not emphasized enough by the authors, in my opinion, is that: in the small-state-space case ($m$ is small), the authors obtain a $poly(n^{O(m)})$ algorithm to find the optimal robust signaling scheme, where $n$ is the number of actions of the receiver.  But by following [Gan et al 2023]'s approach, one can only get a $poly(2^n, n, m)$ exponential-time algorithm because the algorithm needs to enumerate all the $2^n$ subsets of the $n$ actions to check whether they can be induced as a set of $\delta$-best-responding actions.  The authors observe that the number of such feasible subsets cannot be more than $n^{O(m)}$ and design an efficient algorithm (Algorithm 1) to find all of them without enumerating all $2^n$ subsets.  This observation is interesting, and it is a significant improvement over [Gan et al 2023] and a good contribution to the literature.

---

> > > ### Author Response · Authors · 2024-08-14
> > >
> > > Thanks for your support of our paper! We'll add a more detailed discussion of our technical contributions and highlight our results in the small state space setting in revisions of the paper.

---

### Official Review · Reviewer_FR8F · 2024-07-13

**Soundness:** 3
**Presentation:** 2
**Contribution:** 1
**Rating:** 3
**Confidence:** 5

**Summary:**

This paper studies the Bayesian Persuasion problem under the condition that the receiver may respond suboptimally. The authors provide a few computational results on the problem from its computational hardness to the approximation algorithms.

**Strengths:**

The paper considers an important and realistic problem on how to optimize the sender's information design when the receiver may not respond optimally. The paper provides a few computational results on the problem, which is a valuable addition to the existing literature.

**Weaknesses:**

The result of this paper is almost a copy-paste of the paper, Robust Stackelberg Equilibria, by Gan et al. [2023]. Both the hardness result and the design approximation algorithm are identical, and it is unclear what is the technical contribution of this paper, except a slight change of the problem setup. Moreover, it has already been noticed by the community that the Stackelberg game and the Bayesian Persuasion problem share the same structure, and it is not clear what is the new insight that this paper provides. See e.g. [1]. While the paper Gan et al. [2023] is cited, none of the similarities in the technical results are discussed in the paper. This can raise an ethical flag. Hence, I urge the authors to provide a thorough discussion of the technical contribution of this paper and how it is different from the existing literature.

[1] Jiarui Gan, Minbiao Han, Jibang Wu, and Haifeng Xu. Generalized Principal-Agency: Contracts, Information, Games and Beyond.

**Questions:**

Please address my concern in the weakness section.

---

> ### Author Rebuttal · Authors · 2024-08-07
>
> Thank you for your comments.
>
> We respectfully disagree with the view that our paper is "almost a copy-paste" of the work by Gan et al., and that the results are "identical".  While we recognize the high-level similarities between Stackelberg games and Bayesian persuasion, we believe that the community continues to find independent interest in each of these problems. In particular:
>
> **High-level differences between Stackelberg games and Bayesian persuasion**: one could certainly argue that Bayesian persuasion instances are a particular class of Stackelberg games -- in fact, one could make similar arguments about other important fields of algorithmic game theory / economics, such as dominant strategy mechanism design.  However, such a reductionist approach offers little help in understanding the specific structure of Bayesian persuasion even in the most basic settings (see, e.g., "Bayesian Persuasion" by Kamenica and Gentzkow).  When it comes to computation (which is the main focus of our paper), yet another issue arises: the computational complexity of a problem depends crucially on its representation.  The standard representation of an instance of Bayesian persuasion is typically much more succinct than the flat representation of the corresponding Stackelberg game (where the leader's strategy space is all feasible policies of the sender in the Bayesian persuasion instance), which means an "efficient" algorithm under the latter representation may not be "as efficient" under the former representation.
>
> Below we provide a **detailed comparison to the work by Gan et al.**  We will focus on concrete differences, which are easier to describe and verify.  In particular, it should be clear from the differences below that the results are not "identical".
>
> - **Model**: allowing the "agent" / "follower" / "receiver" to choose a response that is suboptimal by a given amount is a standard approach in algorithmic game theory when robustness is desired.  Both our work and that of Gan et al. take this approach.  However, the first major difference (both conceptual and technical) already shows in the respective models: the succinct representation of a Bayesian persuasion instance has the additional component of *states*.  This in particular means the sender's strategy is a randomized mapping from states to posterior beliefs, which is, superficially speaking, of much higher dimensions than a Stackelberg equilibrium.  In fact, since the classical revelation principle is no longer valid, one may suspect the former strategy space is infinite-dimensional (we show this is not the case).  As we argue below, this has significant technical implications to the computation of an (almost) optimal strategy.
>
> - **Structure of optimal strategies**: our positive results rely crucially on a structural property of optimal strategies that we prove (Lemma 3.2), which doesn't have a counterpart in robust Stackelberg games.  The property says that while there are infinitely many possible signals, restricted to optimal strategies, many of them can be grouped, and we only need to consider a finite number of representative signals.  This in particular means the effective strategy space of the sender is finite-dimensional.  Note that here, the receiver may choose different actions depending on the signal sent.  In contrast, the leader's strategy space in a robust Stackelberg game is naturally finite dimensional, and the follower always chooses a fixed action in response to the leader's strategy.
>
> - **Algorithm for fixed number of actions**: our algorithm when the number of actions is fixed (Proposition 3.3) is a natural combination of the classical LP for Bayesian persuasion and the above structural property.  In particular, we solve a single LP for the sender's optimal strategy.  For comparison, the fixed-$n$ algorithm by Gan et al. generalizes the algorithm by Conitzer and Sandholm ("Computing the Optimal Strategy to Commit to"), which enumerates the follower's response and solves one LP for each possibility.
>
> - **Algorithm for fixed number of states**: here we deviate significantly from existing techniques.  In particular, our algorithm relies crucially on the notion of symmetric difference graphs and connectivity therein.  To our knowledge, such techniques have not been employed in the context of Bayesian persuasion or Stackelberg games.  In contrast, there are no states in Stackelberg games to begin with.  Note that the parameter $m$ in Stackelberg games plays an intrinsically different role than the number of states in our model, and Gan et al. present no efficient algorithm when $m$ is fixed (though the comparison itself may not be meaningful in the first place).
>
> - **Hardness result**: first we note that it's not uncommon for a well-motivated problem to be computationally hard, and the fact that both our problem and Gan et al.'s are hard doesn't necessarily mean the two are otherwise similar.  Our hardness result is based on a fundamentally different reduction from Gan et al.'s.  In particular, we reduce from the problem of Subset Sum, whereas Gan et al. reduce from Exact Cover by 3-Sets.  Details of the two reductions bear virtually no similarity.
>
> - **Approximation algorithm**: our algorithm shares the same high-level idea with Gan et al.'s (as well as many other approximation algorithms involving the probability simplex): one first discretizes the probability simplex into a reasonable number of representative points, and then considers the problem restricted to these points.  Despite this high-level similarity, the concrete algorithms are sufficiently different.  Specifically, the two algorithms are based on their exact (and inefficient) versions respectively, which means our algorithm solves a single LP, and Gan et al.'s enumerates the follower's response and solves one LP for each possibility.  Our algorithm also has to deal with additional challenges introduced by the states and the prior distribution.

---

> > ### Comment · Reviewer_FR8F · 2024-08-13
> >
> > I have read the rebuttal as well as other reviews. While I appreciated the detailed comparison to the work by Gan et al. missing in the paper, I am still not convinced that the contribution from this paper is significant enough over Gan et al.. For example, I am not sure whether Lemma 3.2 should be viewed as a positive result (unique to this paper). To me this is an easy result based out of Gan et al.., where the number of \delta-best response regions in robust Stackelberg game is exactly the number of the signals to consider in the robust Bayesian persuasion. I do not think this observation is conceptually new. It is also somewhat unrealistic to consider this many signals in the persuasion problem. The hardness results are also expected (despite a deduction to different hard problems), because function concavification (in bayesian persuasion) is at least as hard as function maximization. In summary, I believe the paper needs to be further improved by explicitly exploring the connections and differences between Stackelberg game and bayesian persuasion problem.

---

> > > ### Author Response · Authors · 2024-08-14
> > >
> > > Thank you for your response.
> > >
> > > Lemma 3.2 is a generalization of the revelation principle to the approximate best response setting. The key distinction between this lemma and the result (Proposition 1) of Gan et al is that, in Stackelberg games, the search space (probability simplex of principal’s strategies) can be directly partitioned into sub-regions in terms of eps-best response sets, whereas in the Bayesian persuasion problem, there is no pre-defined search space since the signal space needs to be defined first. In terms of techniques, to prove that this signal space suffices, our approach requires an iterative proof that begins with any signal space and merges signals without reducing robust utility. This step is unnecessary in the robust Stackelberg games setting.
> > >
> > > We also want to emphasize that the fact that the impracticality of considering this many signals is exactly the motivation for our results with a small state space. We show that one can greatly reduce the number of signals to $n^{O(m)}$ by leveraging the structural insights of how delta response regions correspond to polytopes cut by polynomially many hyperplanes in a low-dimensional space. We further abstract the connectivity of those polytopes into a symmetric different graph, on which we then apply graph algorithms to efficiently search for the $n^{O(m)}$ number of useful signals. In contrast, the $2^n\cdot poly(m,n)$ complexity of Gan et. al’s algorithm does not benefit from this speed-up because the above structure is unique to the Bayesian persuasion setting.

---

### Official Review · Reviewer_BTN6 · 2024-07-24

**Soundness:** 3
**Presentation:** 2
**Contribution:** 2
**Rating:** 5
**Confidence:** 3

**Summary:**

This paper studies Bayesian persuasion settings under approximate best response, where the receiver may choose suboptimal actions based on their beliefs. The authors develop efficient algorithms to compute an (almost) optimal sender commitment. First, they show the failure of the revelation principle. Furthermore, the paper develops polynomial-time exact algorithms for small state or action spaces and a quasi-polynomial-time approximation scheme (QPTAS) for the general problem. It also shows that no polynomial-time exact algorithm exists for the general problem unless P = NP.

**Strengths:**

- The paper studies an interesting problem for the Bayesian persuasion community.
 - The paper shows some interesting results and characterizations.

**Weaknesses:**

- The two main results, QPTAS and the hardness result, are presented at the end of the paper without explanations or intuitions. I believe this aspect should be improved.

- Why, in the 'algorithm with small state spaces', you need the 'explore' algorithm and you introduce the 'Symmetric difference graph'? Maybe I am wrong, so please correct me if I am, but I think you can simply take the vertices of the regions $\Delta_{(A,a)}$ (which are clearly exponential in $m$) and instantiate an LP with those vertices in the space of posterior distributions (see, e.g., Section 2.1 in [1]). This approach would require at most half a page and would simplify the current approach.

- It is unclear to me how different your approach is compared to the one used by [2] when either the number of states or actions is fixed.

- Finally, the knowledge of $\delta>0$ limits the contribution of the work.

[1] Castiglioni, M., Celli, A., Marchesi, A., and Gatti, N. Online bayesian persuasion. Advances in Neural Information Processing Systems, 33, 2020.

[2] Jiarui Gan, Minbiao Han, Jibang Wu, and Haifeng Xu. Robust stackelberg equilibria. In Proceedings of the 24th ACM Conference on Economics and Computation (EC), page 735, 2023.

**Questions:**

- Is the approach discussed above with small state spaces a possible approach?
- Is your approach employable for a multi-type receiver?

---

> ### Author Rebuttal · Authors · 2024-08-07
>
> Thank you for your detailed and insightful comments.
>
> **QPTAS and hardness not explained**: we agree this is suboptimal.  We made a hard choice here due to the strict page limit.  There is a high-level description of the QPTAS in Appendix F.  We find the proof of the hardness result particularly interesting from a technical perspective, though we couldn't find a more concise way to describe the main idea due to the nature of such hardness reductions.  We will use the extra page afforded by the camera-ready version to provide more explanations in the main paper.
>
> **Small state spaces algorithm**: we are not sure if we correctly understand your proposal, but based on what we understand: our algorithm is essentially doing what you proposed.  The "explore" subroutine is needed precisely because we need to *find* the vertices of the regions $\Delta_{(A, \tilde a)}$ -- we don't know which $(A, \tilde a)$ pairs to consider a priori, and we can't consider all of them because just the enumeration would cost too much (time exponential in $n$). Therefore, we abstract the structures of polytopes in the “space of posterior distributions” into the “symmetric difference graph”, on which we can utilize the graph algorithms to efficiently find all feasible signals, i.e., “vertices” that you described. We are happy to continue the discussion if you find the above unsatisfactory.
>
> **Fixed-parameter algorithms, comparison to Gan et al.**: very superficially, our algorithm has to deal with states, while Gan et al.'s doesn't.  In particular, we are not aware of techniques similar to the "explore" subroutine used in the fixed-number-of-states algorithm (given that it is in fact necessary) in similar contexts.  Even our fixed-number-of-actions algorithm combines the classical LP for Bayesian persuasion with the structural property that we prove (Lemma 3.2), whereas Gan et al.'s algorithm generalizes the classical algorithm for Stackelberg equilibrium by Conitzer and Sandholm ("Computing the Optimal Strategy to Commit to").  As a result, our algorithm solves a single LP (which is in a sense necessary), and Gan et al.'s enumerates the follower's response and solves one LP for each possibility.  Please also refer to our response to Reviewer zPak for more detail.
>
> **Knowing $\delta$**: the assumption can be partially relaxed since the sender can estimate $\delta$, e.g., through binary search.  Once a guess of $\delta$ has been made, the sender can implement the corresponding strategy and determine if the guess is too optimistic by observing whether the realized payoff is lower than expected.  Note that here we need certain monotonicity properties: if the guess is too small, then the payoff will be upper bounded by the actual optimal payoff, which is upper bounded by the optimal payoff under the guess; if the guess is too large, then the implemented strategy also works for any smaller $\delta$ (including the actual one), and the resulting payoff will be at least as good.  We will discuss this if space permits.
>
> **Multi-type receiver**: our results should generalize to the case of a fixed number of types.  The idea is to construct the type-free LP for each type separately, and then consider the "product LP" where each product signal is a vector of signals, one for each type.  One would also modify the objective function to incorporate the prior distribution over types.

---

> > ### Comment · Reviewer_BTN6 · 2024-08-13
> >
> > I thank the authors for their responses. After carefully reading all the reviews and rebuttals, I still believe that the paper presents several technical similarities to the one by Gan et al. (2023). This is somewhat expected, given that the two settings share many similarities, but it certainly limits the contribution of the work. For this reason, I will keep my score unchanged.

---

### Decision · Program_Chairs · 2024-09-25

**Decision:**

Accept (poster)

**Comment:**

The paper studies a “robust” version of the classical Bayesian persuasion problem, in which the receiver plays approximate best responses rather than exact ones. Specifically, the paper focuses on the computational aspects of the problem.

One of the main concerns raised by the Reviewers (Reviewers BTN6, FR8F, and zPaK in particular) is about the similarity of the paper with “Robust Stackelberg Equilibria” by Gan et al. [2023]. The latter addresses Stackelberg games in which the follower plays approximate best responses in place of exact ones, in the same vein of the present paper. The concern of the Reviewers is on the technical novelty of the paper compared to [Gan et al., 2023].

Reviewers BTN6 and zPaK, despite concluding that the novelty of some of the results in the paper is somehow limited in the light of [Gan et al., 2023], are generally (weakly) positive about the paper overall. Instead, Reviewer FR8F is largely negative about the paper, claiming that “the result of this paper is almost a copy-paste” of the paper “Robust Stackelberg Equilibria” by Gan et al. [2023]. The claim of Reviewer FR8F is specifically targeted to the hardness result and the quasi-polynomial time approximation scheme (QPTAS) presented in the paper. In support of their claim, Reviewer FR8F is advancing the argument that “it has already been noticed by the community that the Stackelberg game and the Bayesian Persuasion problem share the same structure”.

After reading the paper, the reviews, Authors’ rebuttals, and the following discussion, I developed some considerations that motivate my final decision. I summarize them below.

1. I believe that the claim of Reviewer FR8F on the paper being a “copy-paste” of the work by Gan et al. [2023] is exaggerated. There are indeed some similarities between the works, but these are inevitable given that the present paper is extending the one by Gan et al. [2023] from Stackelberg games to Bayesian persuasion problems. However, I agree with the Reviewers that the authors should have discussed the differences with the work by by Gan et al. [2023] more carefully in their paper.

2. I tend to disagree on the fact that Stackelberg games and Bayesian persuasion problems share the same structure. Bayesian persuasion can indeed be formulated as a particular Stackelberg game, but this completely disrupts the problem representation of Bayesian persuasion and its particular structure. Thus, it is almost always the case that algorithms and results for Stackelberg games are **not** directly applicable to Bayesian persuasion problems. Indeed, if this were the case, then all the research on algorithmic Bayesian persuasion would be pointless, given that we already know almost everything about Stackelberg games.

3. I think that the paper makes nice contributions that completely depart from the results derived by Gan et al. [2023]. As Reviewer R1Q8 is pointing out, one of the most important (and surprising) contributions of the paper is that, in the small-state-space case ($m$ is small), it is possible to obtain a $poly(n^{O(m)})$ algorithm to find an optimal robust signaling scheme. By following the approach in [Gan et al., 2023], one can only get a $poly(2^n, n, m)$ exponential-time algorithm.

In conclusion, given the considerations above and the fact that the paper provides a valuable extension of the model in “Robust Stackelberg Equilibria” by Gan et al. [2023] to the Bayesian persuasion setting, **I recommend that the paper is accepted subject to** the fact that the Authors carefully address Reviewers’ concerns in the final version of the paper. In particular, the authors should provide a detailed discussion of the similarities between the paper and “Robust Stackelberg Equilibria” by Gan et al. [2023].